# Efficacy and safety of next-generation tick transcriptome-derived direct thrombin inhibitors

Cho Yeow Koh [1,14], Norrapat Shih[1,14], Christina Y. C. Yip[2], Aaron Wei Liang Li [1], Weiming Chen[1],
Fathiah S. Amran[1], Esther Jia En Leong[3], Janaki Krishnamoorthy Iyer[3], Grace Croft[1],
Muhammad Ibrahim Bin Mazlan[1], Yen-Lin Chee[4], Eng-Soo Yap[4], Dougald M. Monroe[5], Maureane Hoffman [6],
Richard C. Becker[7], Dominique P. V. de Kleijn[1,8], Vaishali Verma [9], Amita Gupta[9], Vijay K. Chaudhary[9],
A. Mark Richards[10,11], R. Manjunatha Kini [3,12✉] & Mark Y. Chan [1,13✉]

Despite their limitations, unfractionated heparin (UFH) and bivalirudin remain standard-of-care parenteral anticoagulants for percutaneous coronary intervention (PCI). We discovered novel direct thrombin inhibitors (DTIs) from tick salivary transcriptomes and optimised their pharmacologic activity. The most potent, ultravariegin, inhibits thrombin with a $K_i$ of 4.0 pM, 445-fold better than bivalirudin. Unexpectedly, despite their greater antithrombotic effect, variegin/ultravariegin demonstrated less bleeding, achieving a 3-to-7-fold wider therapeutic index in rodent thrombosis and bleeding models. When used in combination with aspirin and ticagrelor in a porcine model, variegin/ultravariegin reduced stent thrombosis compared with antiplatelet therapy alone but achieved a 5-to-7-fold lower bleeding time than UFH/bivalirudin. Moreover, two antibodies screened from a naïve human antibody library effectively reversed the anticoagulant activity of ultravariegin, demonstrating proof-of-principle for antidote reversal. Variegin and ultravariegin are promising translational candidates for next-generation DTIs that may reduce peri-PCI bleeding in the presence of antiplatelet therapy.

[1] Department of Medicine, Yong Loo Lin School of Medicine, National University of Singapore, Singapore, Singapore. [2] Department of Laboratory Medicine, National University Hospital, Singapore, Singapore. [3] Department of Biological Sciences, National University of Singapore, Singapore, Singapore. [4] Department of Haematology, National Cancer Institute, Singapore, Singapore. [5] Division of Hematology/Oncology, University of North Carolina at Chapel Hill, Chapel Hill, NC, USA. [6] Department of Pathology, Duke University, Durham, NC, USA. [7] University of Cincinnati, Cincinnati, OH, USA. [8] Department of Vascular Surgery, University Medical Center Utrecht & Netherlands heart Institute, Utrecht, The Netherlands. [9] Centre for Innovation in Infectious Disease Research, Education, and Training (CIIDRET), University of Delhi South Campus, New Delhi, India. [10] Cardiovascular Research Institute, NUHS, Singapore, Singapore. [11] Christchurch Heart Institute, University of Otago, Otago, New Zealand. [12] Department of Pharmacology, Yong Loo-Lin School of Medicine, National University of Singapore, Singapore, Singapore. [13] Cardiac Department, National University Heart Centre, Singapore, Singapore. [14] These authors contributed equally: Cho Yeow Koh, Norrapat Shih. ✉email: dbskinim@nus.edu.sg; mark_chan@nuhs.edu.sg

During percutaneous coronary intervention (PCI), balloon angioplasty is frequently followed by stent implantation. The procedure causes extensive endothelial disruption and injury, leading to an intense burst of thrombin generation[1–3]. As such, patients are routinely pre-treated with dual antiplatelet therapy (DAPT) and administered an injectable anticoagulant, commonly unfractionated heparin (UFH), during the PCI procedure[3,4]. The limitations of UFH include major bleeding, heparin-induced thrombocytopenia (HIT) and the need for coagulation monitoring due to its unpredictable pharmacokinetics. However, UFH remains one of the most widely used parenteral anticoagulants due to low cost (~$4–$10 USD per PCI[5]) and the wealth of clinical experience accumulated in more than eight decades of use[6]. To overcome some of UFH's disadvantages, an injectable direct thrombin inhibitor (DTI), bivalirudin, was developed as an alternative to UFH in PCI[3,4,7]. Although bivalirudin is more expensive (~$400 to $600 per PCI without post-procedural infusion[5]), initial randomised trials showed that bivalirudin was associated with similar antithrombotic efficacy but less bleeding when compared with a combination of UFH and a platelet glycoprotein IIb/IIIa inhibitor[8–10]. However, protocol-mandated use of glycoprotein IIb/IIIa inhibitors in the UFH arm of these trials could have contributed to the higher bleeding rates among patients randomised to UFH. More recent trials with balanced use of glycoprotein IIb/IIIa inhibitors in both bivalirudin and UFH arms have shown less favourable results for bivalirudin[11–14]. Therefore, in the 2018 European Society of Cardiology/European Association for Cardio-Thoracic Surgery guidelines on myocardial revascularization, routine use of UFH received a higher recommendation (class I) than bivalirudin (class IIb) for peri-PCI anticoagulation[15]. Bivalirudin has been more widely adopted in the United States than in the rest of the world. One study estimated that 47.1% of PCI cases between July 2009 and December 2014 in the US used bivalirudin (52.9% UFH) but also noted a decline in bivalirudin use after 2013[16]. With increasing use of potent platelet P2Y$_{12}$ antagonists such as ticagrelor, prasugrel and cangrelor, there remains an even greater unmet is a need for safer peri-PCI anticoagulants to adequately improve the efficacy-safety balance of antithrombotic therapy during PCI[7,17]. Combination antiplatelet and anticoagulant therapy has become more common especially among patients with atrial fibrillation or venous thromboembolism undergoing PCI[15,18].

Haematophagous animals such as leeches, mosquitoes, ticks, tsetse flies and others are rich sources of antithrombotic agents[19]. Molecules such as hirudin from the medicinal leech[20], anophelin from mosquitoes[21,22], madanin from ticks[23] and tsetse thrombin inhibitor from tsetse flies[24,25] are potent and selective thrombin inhibitors that are highly amenable to customisation as synthetic inhibitors with improved properties[25–28]. We have previously characterised a new class of DTIs, variegin and other variegin-like peptides, from the salivary gland of the tropical bont tick (*Amblyomma variegatum*)[29,30]. Variegin, a 32-residue peptide, binds to both thrombin's active site and exosite-I. Variegin inhibits thrombin more potently than bivalirudin by sixfold and demonstrates superior selectivity for thrombin[29,31,32]. Bivalirudin has a half-life of 25 min[8] and must therefore be administered as a continuous infusion during PCI as the majority of these procedures last ~30 min to an hour. Variegin has a half-life of 52.3 ± 4.4 min[33] and may potentially be given as a single bolus for peri-PCI anticoagulation.

In this study, we hypothesise that a high-affinity, high-specificity bivalent DTI with non-covalent binding to thrombin like variegin would sufficiently prevent thrombus formation during PCI when administered in low doses in combination with aspirin and ticagrelor (DAPT); this low-dose approach to peri-PCI anticoagulation

would then preserve the ability to regenerate thrombin when bleeding occurred. Through iterative design and optimisation, we develop a picomolar affinity DTI, named ultravariegin. We perform global coagulation experiments in plasma to better understand the antithrombotic efficacy versus preservation of haemostatic capacity by taking into consideration the interaction of DTIs with platelets and DAPT. In dose-ranging studies using rodent models of carotid artery thrombosis and tail bleeding, we compare the therapeutic indices of variegin and ultravariegin with UFH and bivalirudin. Then, in an ex vivo porcine model of coronary stent thrombosis and venous bleeding, we test the efficacy and safety of these anticoagulants in the absence and presence of DAPT. Finally, we identify specific antidotes for ultravariegin through a screen against a naïve human antibody library and test them against ultravariegin in vitro and in vivo.

## Results

### Design of ultravariegin, a picomolar thrombin inhibitor, from the tropical bont tick *A. variegatum*. 
Variegin-like thrombin inhibitors in Amblyomminae are synthesised as larger precursor proteins containing multiple repeats that are post-translationally processed into shorter peptides[30]. Potential thrombin inhibitor precursors in *A. variegatum* were identified from its salivary gland transcriptome[34] (Fig. 1). Proteomic data and sequence alignment suggest that these peptides have serine residue at the amino terminus[29,30]. However, the post-translational cleavage site remains unidentified[29,30]. Considering the variability in sequences, there is uncertainty about the identity of carboxyl-terminal amino acids of these inhibitors (Fig. 1). We synthesised three peptides representing repeats from one of the precursors (Gen-Bank accession number DAA34688.1 repeat 1, 1B and 1C) with different possible C-termini (Supplementary Figs. 1–14 and Supplementary Table 1). These peptides potently inhibited thrombin's amidolytic activity on chromogenic substrate S2238 with significant inhibition at equimolar concentrations of inhibitor and thrombin with linear inhibition progress curves (Fig. 2a). This is consistent with the previously reported fast- and tight-binding behaviour of variegin-like peptides[29,30]. Truncation of amino acids from the C-terminus resulted in progressively stronger inhibition as indicated by the corresponding inhibitory constant ($K_i$) values (Table 1). The $K_i$ value for DAA34688.1 repeat 1C is 6.5 ± 1.3 pM, which is 42-fold and 96-fold stronger than synthetic variegin and avathrin, respectively (Table 1). We hypothesised that mutating the non-conserved Thr22 in DAA34688.1 repeat 1C to Glu would enrich negative charges needed for binding to thrombin exosite-I. Consistent with this design, the $K_i$ improved to 4.0 ± 0.5 pM (Table 1 and Fig. 2b). We named this mutant ultravariegin, which has a $K_i$ for thrombin that is 445-fold greater than bivalirudin (Table 1). The apparent $K_i$ ($K_i'$) of ultravariegin increased linearly with increasing concentration of thrombin chromogenic substrate, indicating that ultravariegin was a competitive inhibitor of thrombin with respect to thrombin's active site (Fig. 2c).

### Novel sequences located at the C-terminal of ultravariegin impart high affinity towards thrombin. 
As reported earlier, the N-terminal segment of variegin (residues 1–7) interacts with thrombin through long-range electrostatic steering. Residues 8–14 of variegin target thrombin's active site and the C-terminal segment binds to thrombin's exosite-I[29–32]. Thrombin-inhibiting peptides found in *A. variegatum* (Fig. 1) show the highest sequence variability within the C-terminal region. In UV003, UV004 and UV005, we substituted ultravariegin residues in segments to the sequence of variegin to investigate which local segment of ultravariegin is most likely responsible for improved

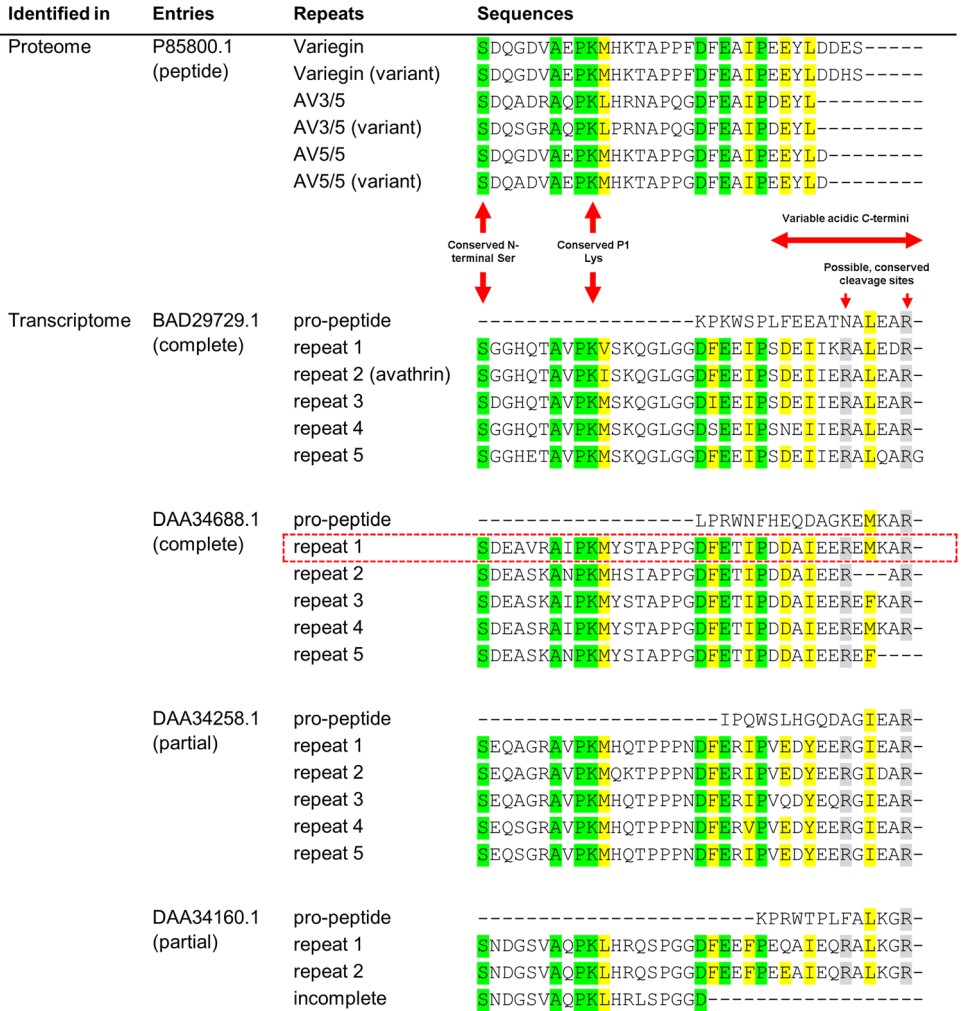

**Fig. 1 Thrombin-inhibiting peptides from *A. variegatum*.** Precursor transcripts are indicated by GenBank accession numbers. Each row within the transcripts represent repeats that may be processed into mature thrombin-inhibiting peptides. Identical and highly similar amino acid residues were highlighted in green and yellow, respectively. Potential cleavage sites are highlighted in grey. The sequence representative of repeat 1 of DAA34688.1 (red dotted box) was synthesised for investigation.

activity over variegin. Substitutions in the C-terminal segment, predicted to target thrombin's exosite-I, is most intolerable. $K_i$ of ultravariegin increased fourfold to 16 pM in UV005 (Table 1). In contrast, replacement of the N-terminal segment (UV003) and thrombin's active site binding segment (UV004) in ultravariegin with corresponding sequences of variegin resulted in minimum changes in $K_i$. The sensitivity of ultravariegin activity to changes in C-terminal residues was also demonstrated by UV012, whereby the A27E mutation increased the $K_i$ > fivefold. Overall, ultravariegin was the most potent peptide (Table 1 and Fig. 2b).

We have previously reported that thrombin cleaves variegin between the Lys10-His11 scissile bond but the cleavage product C-terminal to the scissile bond (sequence: MHKTAPPFDFEAI-PEEYLDDES) is a non-competitive inhibitor of thrombin's active site with a $K_i$ of ~14.1 nM[31]. We synthesised the equivalent cleavage product of ultravariegin UV011 and showed that it inhibits thrombin (Table 1 and Fig. 2d). Assuming the same non-competitive mode of inhibition, the $K_i$ of UV011 is $1.66 \pm 0.76$ nM. Bivalirudin is also cleaved by thrombin upon binding[31,35] and we synthesised the equivalent cleavage product of bivalirudin (BV001) (Table 1). Surprisingly, BV001 was not only unable to inhibit thrombin but instead activate thrombin's active site by up to 20% when tested at concentrations higher than 1 μM (Fig. 2d).

Ultravariegin was then screened for selectivity against 11 serine proteases involved in blood coagulation and fibrinolysis, as well as trypsin and chymotrypsin. At 0.1 nM, ultravariegin inhibited thrombin at around 27%, whereas even at 100 μM, ultravariegin did not inhibit any other serine protein by more than 20%, indicating at least 1,000,000-fold selectivity in preference for thrombin over other serine proteases (Fig. 2e). Thus, ultravariegin appeared to be a viable lead molecule with a substantially optimised $K_i$ at 4 pM and enhanced selectivity for thrombin.

**Haemostatic capacity of blood is largely preserved with variegin or ultravariegin in combination with DAPT.** We then compared variegin and ultravariegin with UFH and bivalirudin. The activated partial thromboplastin time (aPTT) was dose-dependently prolonged by all four compounds (Fig. 3a). Molar estimates of the concentrations of UFH used were derived to allow for comparisons with the other three compounds on the same axis. Among the three DTIs, ultravariegin appeared to be most potent, followed by variegin and bivalirudin, consistent with their respective affinity towards thrombin (Table 1). We then performed clot waveform analysis (CWA), in which successive derivatives of the clotting curves indicate activity of individual coagulation enzymes or complexes. The first derivative (min1) represents thrombin activity (i.e., thrombin burst) and bleeding

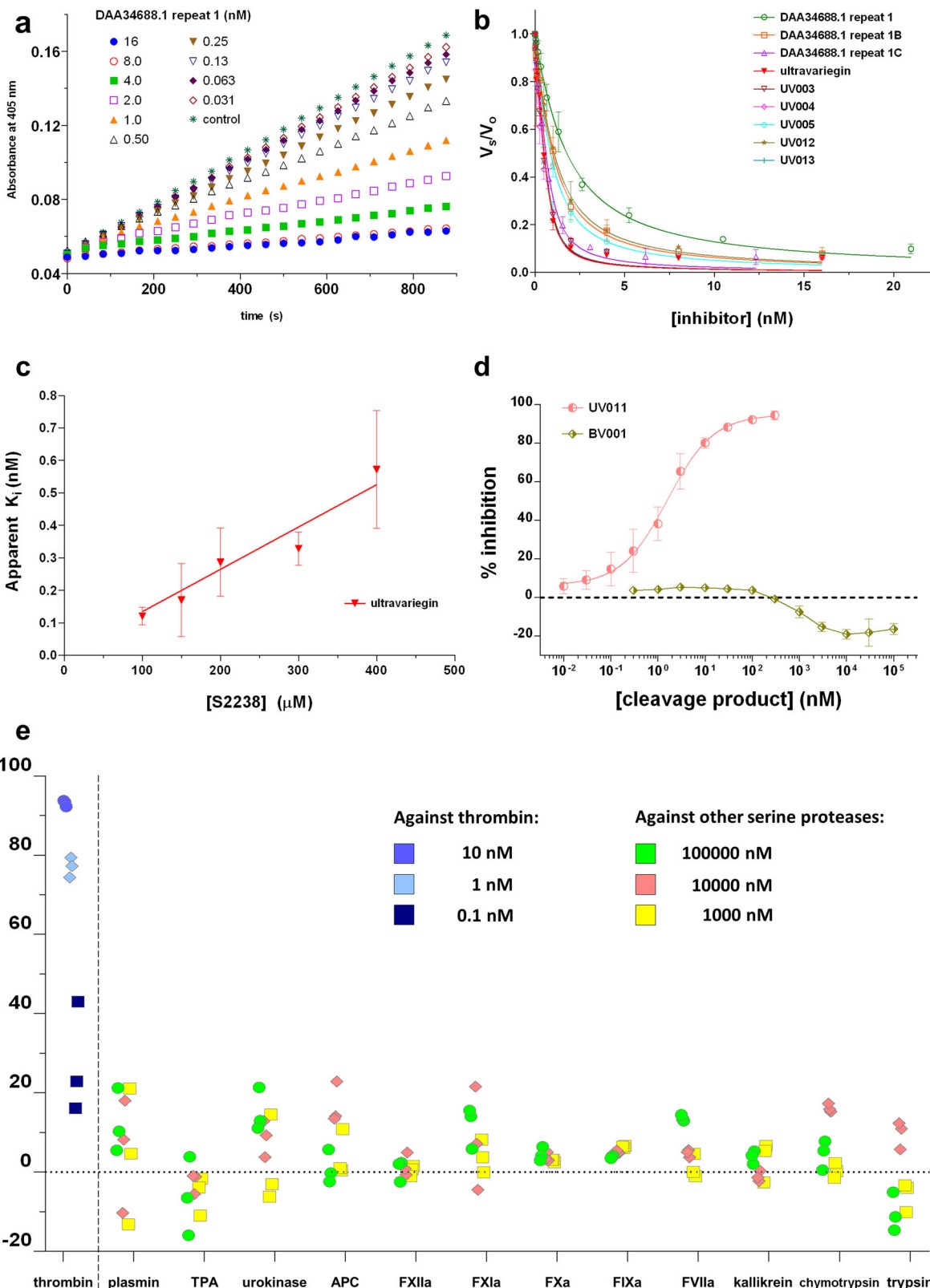

risk (low absolute value is associated with greater bleeding risk)[36,37], the second derivative (min2) represents the activity of the prothrombinase complex and the third derivative (min3) represents activity of the tenase complex[38]. All three derivatives dose-dependently decreased with each of the four anticoagulants tested. However, the rates declined very sharply above 25 nM of

UFH while a decline only occurred when variegin, ultravariegin and bivalirudin were above 1 µM (Fig. 3b).

In the thrombin-generation test (TGT), lag-time (LT) and time-to-peak (TTpeak) for thrombin generation generally increased with increasing concentrations of all four anticoagulants in platelet-poor plasma (PPP) or platelet-rich plasma

**Fig. 2 Inhibition of thrombin by DTIs. a** Amidolytic activity of thrombin (0.8 nM) on chromogenic substrate (S2238, 100 μM) in the presence of various concentrations of inhibitors were monitored as an increase in absorbance over time. A representative progression curve of thrombin inhibition by DAA34688.1 repeat 1 is depicted. All experiments were repeated independently as indicated below with similar results. The linear progression curves are characteristic of fast-binding inhibitors. **b** Residual thrombin amidolytic activity in the presence of various concentrations of various peptides were fitted to a kinetic equation describing tight-binding inhibitors to estimate apparent $K_i$ ($K_i'$), $n = 3$. **c** Ultravariegin showed a linear increase of $K_i'$ with increasing concentrations of substrate [100 μM ($n = 7$); 150 μM ($n = 6$); 200 μM ($n = 6$); 300 μM ($n = 5$); 400 μM ($n = 6$)], indicating competitive inhibition. The $K_i$ were calculated to be 4.0 ± 0.5 pM. **d** Despite cleavage by thrombin, the cleaved peptide C-terminus to scissile bond for ultravariegin (UV011, coloured salmon) retained strong inhibition against thrombin amidolytic activity with $IC_{50} = 1.66 ± 0.76$ nM ($n = 3$). In contrast, the cleaved peptide C-terminus to scissile bond for bivalirudin (BV001, coloured olive) does not inhibit thrombin but instead paradoxically activates thrombin amidolytic activity by around 20% at high concentrations (1–100 μM, $n = 4$). **e** Inhibition of the amidolytic activity of various serine proteases by ultravariegin ($n = 3$), note the difference in peptide concentrations tested against thrombin compared to other serine proteases. All data are mean ± standard deviation (SD), $n$ is number of independent experiments.

| **Table 1 Inhibitory constant ($K_i$) of thrombin-inhibiting peptides.** | | |
|---|---|---|
| **Peptide** | **Sequence** | **$K_i$ (pM)** |
| Bivalirudin | FPRPGGGG–NGDFEEIPEEYL | 1780 ± 152 |
| BV001 | PGGGG–NGDFEEIPEEYL | No inhibition; activation at >1 μM |
| Variegin | SDQGDVAEPKMHKTAPPFDFEAIPEEYLDDES | 277 ± 56 |
| Avathrin | SGGHQTAVPKISKQGLGGDFEEIPSDEIIE | 624 ± 86 |
| DAA34688.1 repeat 1 | SDEAVRAIPKMYSTAPPGDFETIPDDAIEEREMKAR | 43.2 ± 8.0 |
| DAA34688.1 repeat 1B | SDEAVRAIPKMYSTAPPGDFETIPDDAIEER | 20.5 ± 4.7 |
| DAA34688.1 repeat 1C | SDEAVRAIPKMYSTAPPGDFETIPDDAIEE | 6.5 ± 1.3 |
| Ultravariegin | SDEAVRAIPKMYSTAPPGDFEEIPDDAIEE | 4.0 ± 0.5 |
| UV003 | SDQGDVAIPKMYSTAPPGDFEEIPDDAIEE | 4.2 ± 1.0 |
| UV004 | SDEAVRAEPKMHKTAPPGDFEEIPDDAIEE | 4.6 ± 0.4 |
| UV005 | SDEAVRAIPKMYSTAPPGDFEEIPEEYLDDES | 16.0 ± 0.3 |
| UV012 | SDEAVRAIPKMYSTAPPGDFEEIPDDEIEE | 23.0 ± 8.1 |
| UV013 | SDEAVRAIPKMYSQAPPGDFEEIPDDAIEE | 4.5 ± 1.6 |
| UV011 | MYSTAPPGDFEEIPDDAIEE | 1660 ± 759 |

All $K_i$ values were determined through inhibition of thrombin amidolytic activity on chromogenic substrate S2238. Values shown are mean ± standard deviation (SD), $n = 3$ for all peptides except ultravariegin, in which $K_i$ were determined as described in Fig. 2c. In bivalirudin sequence, F represent D-Phe.

(PRP), with or without DAPT (Supplementary Fig. 15). However, key differences were observed in the rate of increase in LT and TTpeak with increasing concentrations of anticoagulants. The dose–response changes of variegin and ultravariegin were gradual compared with UFH and bivalirudin (Fig. 3c and Supplementary Fig. 16). Across all conditions, gradients of LT for variegin and ultravariegin were 1.9- to 6.4-fold lower than for UFH and bivalirudin; gradients of TTpeak for variegin and ultravariegin were 2.3- to 13.7-fold lower than for UFH and bivalirudin (Supplementary Table 2). Relative to UFH and bivalirudin, increases in variegin and ultravariegin concentration result in a more moderate inhibition of thrombin by gradually delaying the thrombin burst.

Endogenous thrombin potential (ETP) for variegin, ultravariegin and bivalirudin were largely maintained around the baseline, indicating the capacity to generate thrombin was not fully abrogated by the three DTIs (Supplementary Fig. 16). The peak thrombin and velocity index (VI) were increased for variegin across a 100-fold concentration range (0.01–1 μM) in PPP and PRP without DAPT, but this increase was attenuated in PRP with DAPT (Fig. 3d, Supplementary Fig. 16 and Supplementary Table 2). For ultravariegin, the thrombin peak and VI showed the most gradual change across a 1000-fold concentration range (0.001–1 μM) in PPP and PRP (Fig. 3d, Supplementary Fig. 16 and Supplementary Table 2). In contrast, ETP, peak thrombin and VI were severely impaired in plasma spiked with UFH, indicating inhibition of thrombin production that did not recover within the timeframe of the experiment (Supplementary Fig. 16). The dose–response curves for UFH were steep, with near-complete inhibition achieved within the small (eightfold)

concentration range tested (0.025–0.2 μM) (Fig. 3d, Supplementary Fig. 16 and Supplementary Table 2). ETP, peak thrombin and VI increased within the small dose range (fourfold) of bivalirudin tested (1.5–6 μM) under all conditions, suggesting a potential risk for excessive thrombin generation on the rebound (Fig. 3d, Supplementary Fig. 16 and Supplementary Table 2). Taken together, these results suggest that thrombin-generation capacity for haemostasis was best preserved when ultravariegin, and to a lesser extent, variegin, was used in combination with DAPT.

**Variegin and ultravariegin showed a wider therapeutic index than UFH and bivalirudin in a rodent model.** In a rat model of FeCl$_3$-induced carotid artery thrombosis, the time taken for complete carotid artery occlusion increased dose-dependently with all four anticoagulants. A single 5 mg/kg IV bolus injection of variegin resulted in an occlusion time close to 60 min (maximum observed duration) (Fig. 4a). Consistent with its lower $K_i$ value for thrombin inhibition, a lower dose of ultravariegin at 2 mg/kg fully prevented carotid artery occlusion (Fig. 4b). This maximum level of antithrombotic efficacy was similarly observed with clinically approved doses of UFH and bivalirudin. For UFH, this therapeutic dose was a bolus IV injection of 432 U/kg which translates to a human equivalent dose of 70 U/kg (Fig. 4c). For bivalirudin, it was 10.8 mg/kg/h of IV infusion, which translates to a human equivalent dose of 1.75 mg/kg/h (Fig. 4d).

In the tail bleeding model, bleeding time increased dose-dependently for all four compounds. The occlusion time and bleeding time curves for variegin and ultravariegin were well-separated, suggesting a wider therapeutic index (TI) than either

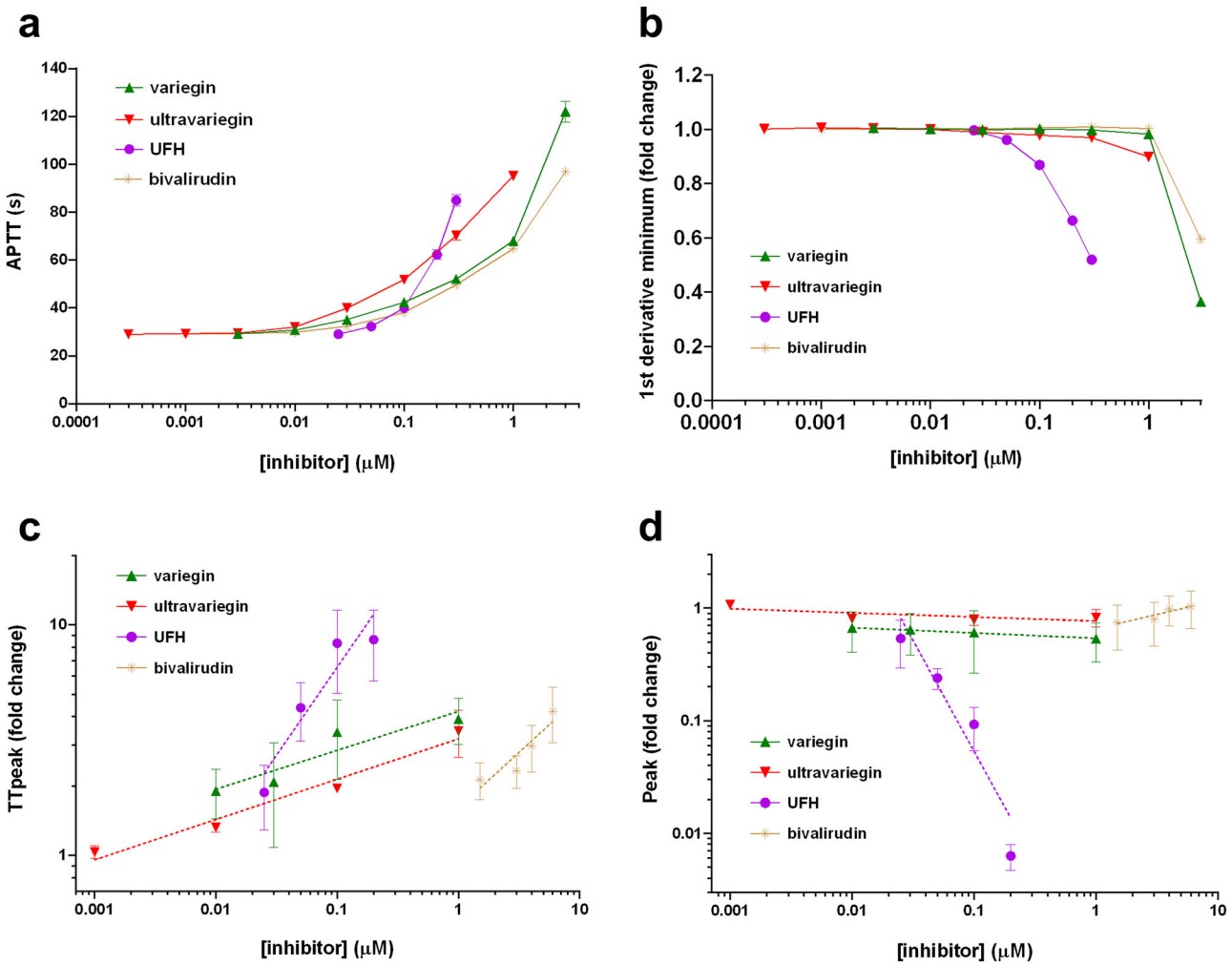

**Fig. 3 Effects of variegin, ultravariegin, UFH and bivalirudin in human plasma. a** APTT using PPP. **b** Minimum values from first derivative curves using PPP. **c** Time-to-peak (TTpeak) and (**d**) peak thrombin concentration were plotted as fold change over vehicle controls in double logarithmic plots. The slopes of the plots indicate how fast anticoagulant intensity changed with dose and is listed in Supplementary Table 2. Experiments shown in (**c**) and (**d**) were performed in PRP in the presence of DAPT. Plots for other parameters including lag-time, endogenous thrombin potential (ETP) and velocity index (VI) can be found in Supplementary Fig. 16. Data shown are mean ± SD, $n = 3$ biologically independent samples.

UFH or bivalirudin. With the estimated therapeutic dose of variegin (5 mg/kg) and ultravariegin (2 mg/kg), the bleeding time was only at around half of the maximum bleeding time (~30 min). In contrast, the therapeutic dose of UFH and bivalirudin resulted in maximal bleeding (Fig. 4c, d). The dose–response curves for antithrombotic efficacy (occlusion time) and safety (bleeding time) largely overlapped for UFH, indicating a very low therapeutic index, while the same set of curves for bivalirudin showed minor separation. We then estimated the dose of each compound that resulted in a 50% maximum response time (RT50; 30 min) for occlusion time and bleeding time experiments (Fig. 4). The TIs for each compound were calculated as the dose at RT50 for bleeding time divided by occlusion time. The TIs for UFH and bivalirudin were 1.0 and 1.3, respectively, while the TIs for variegin and ultravariegin were 3.8 and 6.7, respectively, around three- to sevenfold better than UFH and bivalirudin (Supplementary Table 3). The wider TIs observed with variegin and ultravariegin are consistent with the preceding in vitro human blood experiments in which inhibition of coagulation is effective but gradual, and haemostatic capacity is preserved.

We further validated the results of the tail bleeding model with a different rodent bleeding model, the saphenous vein bleeding

model[39,40]. We tested variegin, ultravariegin, UFH and bivalirudin at their respective doses that resulted in a 50% maximum response time of occlusion in the carotid artery thrombosis model (RT50; 30 min). In the saphenous vein bleeding model, a higher number of haemostatic events indicate better preservation of haemostatic capacity[39,40]. We showed that the number of haemostatic events ranked as follow: ultravariegin > variegin > bivalirudin > UFH (Fig. 4e). The average time per bleeding event had an identical rank order as tail bleeding time experiments: UFH > bivalirudin > variegin > ultravariegin (Fig. 4f). The bleeding time from both models showed strong, positive correlation ($r = 0.978$) (Fig. 4f, insert), indicating consistent results from both bleeding models.

**Variegin or ultravariegin is more effective in preventing stent thrombosis than UFH and bivalirudin.** We then compared the efficacy of the four anticoagulants in an ex vivo porcine model of stent thrombosis in the absence and presence of DAPT (Supplementary Fig. 17). This model investigates thrombosis when blood is exposed to thrombogenic tissue under flow conditions, and has been commonly employed for antithrombotic drug testing[41–45]. Simultaneously, we compared the safety of these compounds by performing bleeding time experiments on a

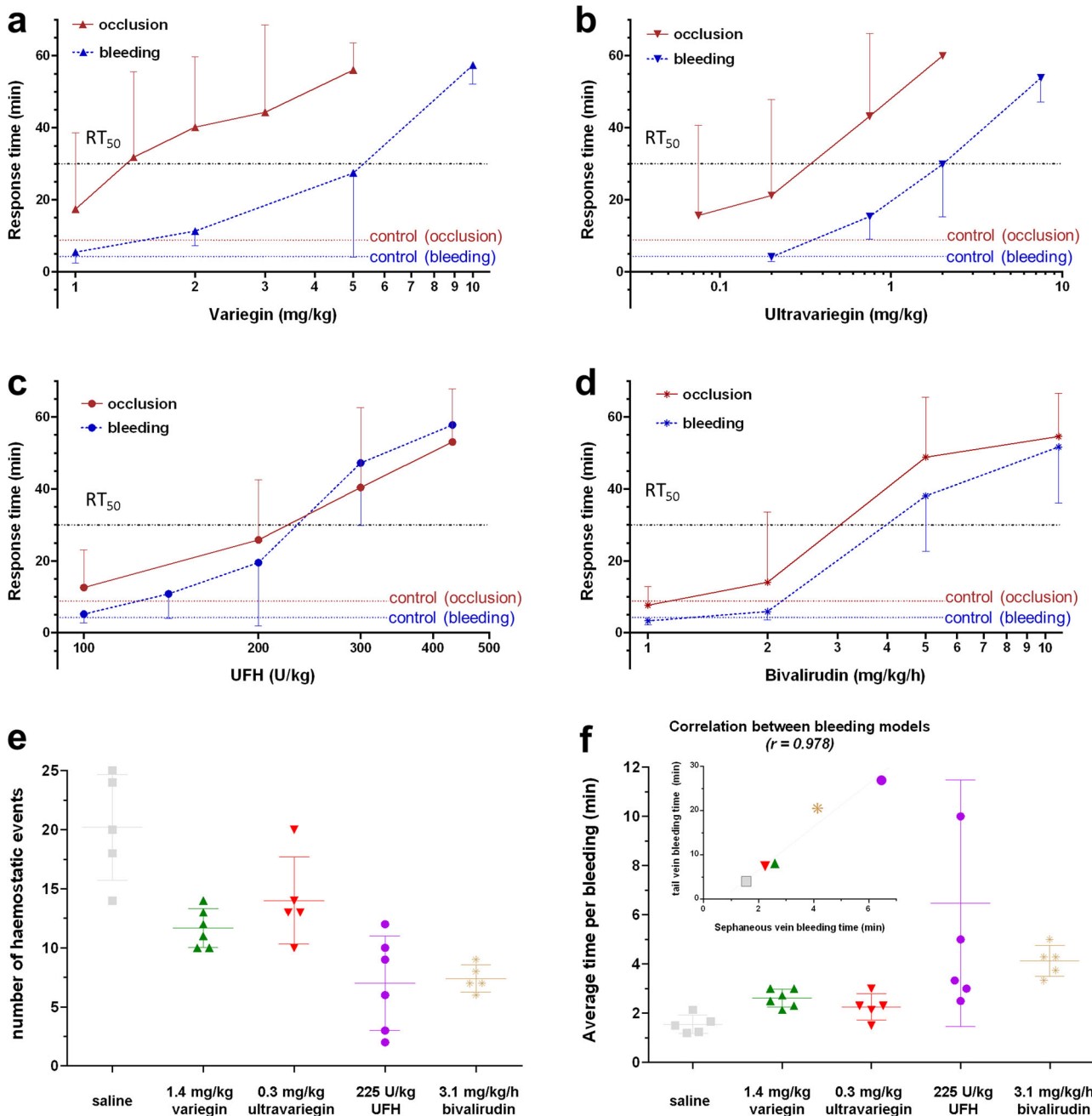

**Fig. 4 Efficacy-safety profiles of anticoagulants in rat models.** Efficacy and safety were determined using $FeCl_3$-induced carotid artery thrombosis and tail incision bleeding models, respectively. The duration of observations (y axis) for both models were standardised at 60 min and titled as Response Time. This facilitates the visualisation of the separation between efficacy (occlusion, red) and safety (bleeding, blue) at any given dose of anticoagulants. Time taken for the carotid artery to be completely occluded (red) was plotted against increasing doses of (**a**) variegin at 1.0 (n = 6), 1.4 (n = 8), 2.0 (n = 8), 3.0 (n = 6), 5.0 (n = 7) mg/kg; **b** ultravariegin at 0.075 (n = 5), 0.2 (n = 7), 0.75 (n = 5), 2.0 (n = 3) mg/kg; **c** UFH at 100 (n = 5), 200 (n = 5), 300 (n = 6), 432 (n = 8) U/kg and **d** bivalirudin at 1.0 (n = 6), 2.0 (n = 7), 5.0 (n = 8), 10.8 (n = 5) mg/kg/h. Time for occlusion in rats injected with saline (red dash line) was 8.82 ± 3.57 min (n = 6). Time taken for bleeding to stop (blue), was plotted against increasing doses of (**a**) variegin at 1.0 (n = 6), 2.0 (n = 7), 5.0 (n = 8), 10 (n = 7) mg/kg; **b** ultravariegin at 0.2 (n = 5), 0.75 (n = 6), 2.0 (n = 5), 7.5 (n = 5) mg/kg; **c** UFH at 100 (n = 5), 140 (n = 6), 200 (n = 8), 300 (n = 7), 432 (n = 5) U/kg; and **d** bivalirudin at 1.0 (n = 6), 2.0 (n = 7), 5.0 (n = 7), 10.8 (n = 6) mg/kg/h. Time for bleeding in rats injected with saline (blue dash line) was 4.25 ± 1.67 min (n = 5). Doses of respective anticoagulants to elicit 50% of response in the models (RT_{50}, dotted lines) were estimated and used to calculate the therapeutic index (TI) as listed in Supplementary Table 3. **e** The number of haemostatic events (i.e., clot formation) and **f** average time per bleeding event (i.e., bleeding time) within 30 min in the saphenous vein bleeding model were used to validate results from tail incision bleeding model at doses in which efficacy for respective treatments are at 50% (ie RT_{50} for occlusion): saline control (n = 5), 1.4 mg/kg variegin (n = 6), 0.3 mg/kg ultravariegin (n = 5), 225 U/kg UFH (n = 6), 3.1 mg/kg/h bivalirudin (n = 5). Figure insert shows correlation coefficient r = 0.978 (p = 0.004) between bleeding time obtained from tail incision model (y axis) and sapheneous vein bleeding model (x axis). Data shown are mean ± SD.

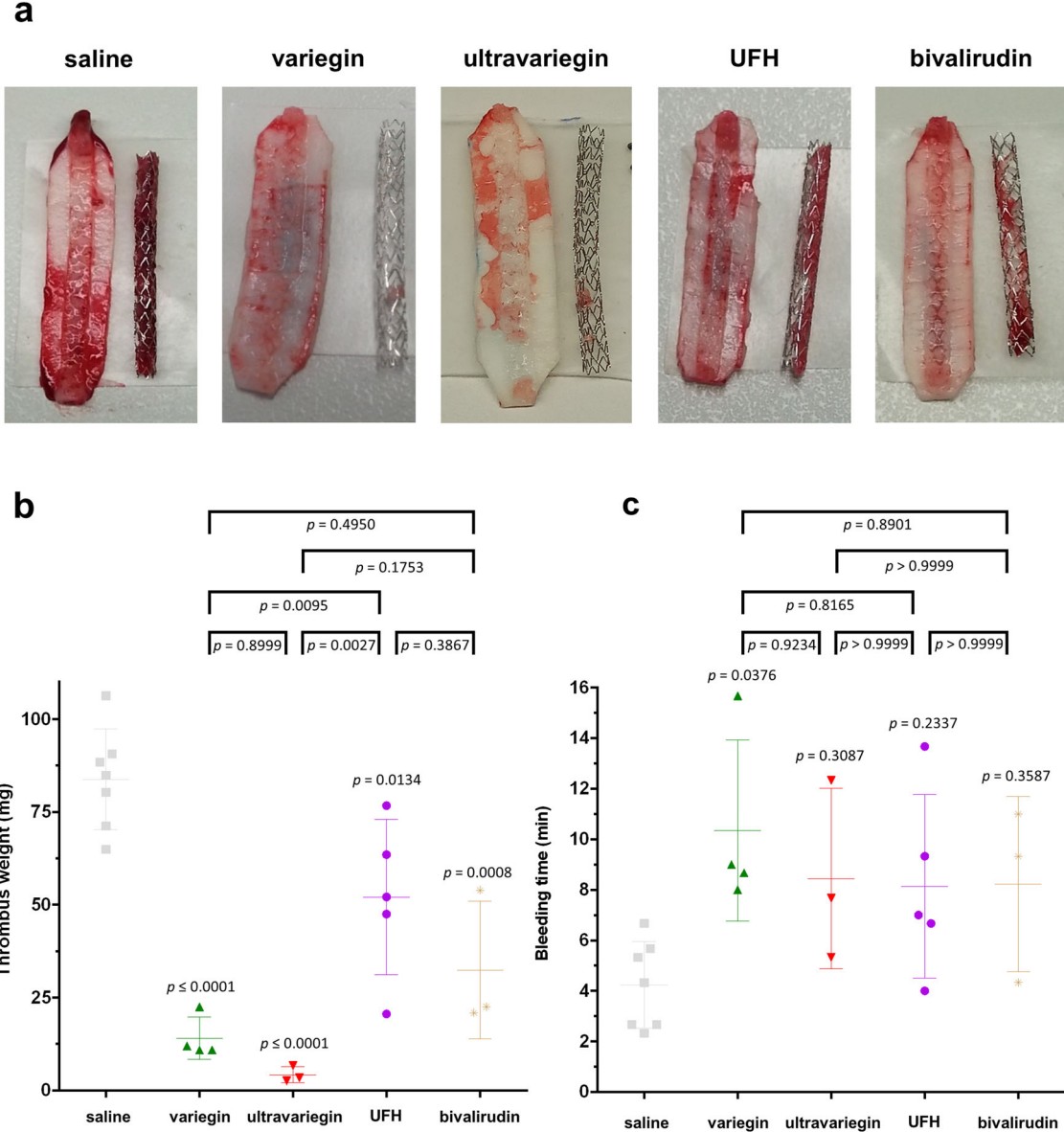

**Fig. 5 Efficacy-safety profiles of anticoagulants (without DAPT) in porcine models.** Efficacy and safety were determined using ex vivo stent thrombosis and superficial ear vein bleeding models, respectively. **a** Representative photographs showing thrombi formed on coronary stents and endothelium-denuded pig aorta strips in pigs administered with saline (i.v. bolus, $n = 7$), 1 mg/kg variegin (i.v. bolus, $n = 4$), 0.25 mg/kg ultravariegin (i.v. bolus, $n = 3$), 100 u/kg UFH (i.v. bolus, $n = 5$) and 0.75 mg/kg (i.v. bolus) plus 1.75 mg/kg/h (i.v. continuous infusion) bivalirudin ($n = 3$). Experiments were repeated in the indicated number of animals independently with similar results. **b** Total weight of thrombi (one-way ANOVA $P < 0.0001$). **c** Bleeding time (one-way ANOVA $P = 0.0474$). Multiplicity-adjusted $P$ values for post hoc Tukey's multiple comparisons between respective anticoagulants with saline were indicated immediately above each treatment. Multiple comparisons among anticoagulants were indicated on top of the plot. Data shown are mean ± SD.

superficial ear vein of the same animals. Without anticoagulation (saline), the stents were completely occluded with thrombus within 60 min of blood flow through the extracorporeal circuit (Fig. 5a, b). With 1 mg/kg of variegin, the therapeutic dose translated based on rodent experiments, thrombus formation was reduced by ~83% (Fig. 5a, b). Near-complete thrombus reduction was achieved with pigs receiving 0.25 mg/kg of ultravariegin (Fig. 5a, b). The administration of therapeutic doses of UFH and bivalirudin reduced thrombus formation by ~38% and 61%, respectively (Fig. 5a, b). Compared with saline, all anticoagulants increased bleeding time by 1.9- to 2.4-fold although only the difference between variegin and saline achieved statistical significance (4.2 min vs 10.3 min, multiplicity-adjusted $P$ value = 0.0376) (Fig. 5c).

**Low doses of variegin or ultravariegin in combination with DAPT are highly efficacious in preventing stent thrombus formation without increasing bleeding time.** When DAPT (300 mg aspirin and 180 mg ticagrelor loading dose) was orally administered 16 h prior to the experiments, in the absence of anticoagulant, thrombus formation on both the aortic strip and stents was reduced significantly compared with saline (10.8 g vs 83.8 mg, two-tailed Student's $t$ test $P \leq 0.0001$), implying that DAPT was partially effective in preventing PCI-related stent thrombosis (Figs. 5a, b and 6a, b). Given the high affinity of variegin and ultravariegin for thrombin, we hypothesised that their combination with DAPT should provide enough antithrombotic efficacy at a much reduced dose of either variegin or ultravariegin, thereby avoiding excessive inhibition of haemostasis. We, therefore, reduced

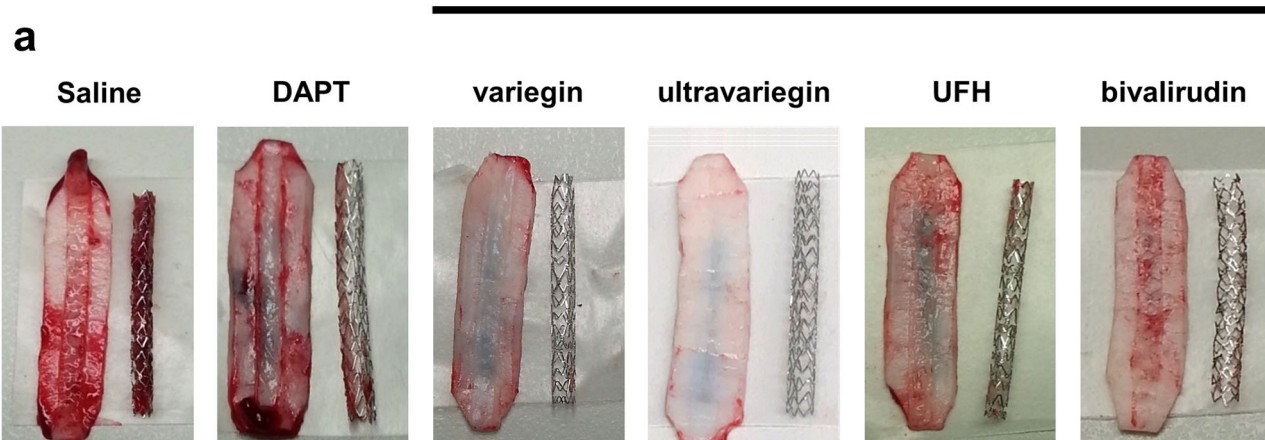

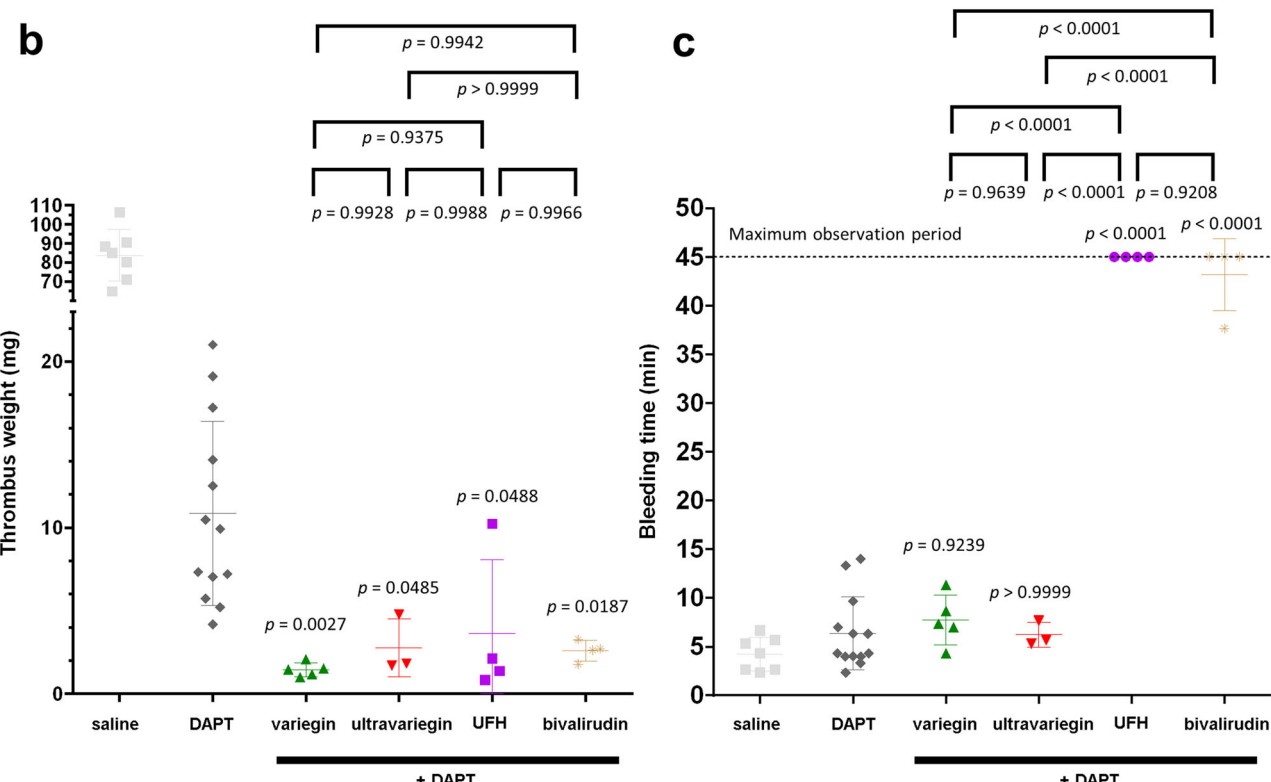

**Fig. 6 Efficacy-safety profiles of anticoagulants (with DAPT) in porcine models.** Efficacy and safety were determined using ex vivo stent thrombosis and superficial ear vein bleeding models, respectively. All pigs were administered 300 mg aspirin and 180 mg ticagrelor orally (DAPT regimen) 16 h prior to the experiments. **a** Representative photographs showing thrombi formed on coronary stents and endothelium-denuded pig aorta strips in pigs administered with DAPT only ($n = 13$), DAPT with 0.1 mg/kg variegin (i.v. bolus, $n = 5$), DAPT with 0.025 mg/kg ultravariegin (i.v. bolus, $n = 3$), DAPT with 100 u/kg UFH (i.v. bolus, $n = 4$), and DAPT with 0.75 mg/kg (i.v. bolus) plus 1.75 mg/kg/h (i.v. continuous infusion) bivalirudin ($n = 4$). Photograph from a pig receiving saline without DAPT, as depicted in Fig. 5a, is reproduced here for comparison. Experiments were repeated the indicated $n$ number of animals independently with similar results. **b** Total weight of thrombi (one-way ANOVA $P = 0.0007$). **c** Bleeding time (one-way ANOVA $P < 0.0001$). Multiplicity-adjusted $P$ values for post hoc Tukey's multiple comparisons between respective anticoagulants with DAPT are indicated immediately above each treatment. Multiple comparison among anticoagulants were indicated on top of the plot. Data shown are mean ± SD.

the dose of variegin and ultravariegin by 10-fold to 0.1 mg/kg and 0.025 mg/kg, respectively, for pigs pre-treated with DAPT. Consistent with our hypothesis, the low dose of variegin and ultravariegin reduced thrombus formation from DAPT alone by 87% and 75%, respectively (Fig. 6a, b). In comparison, the clinically approved doses of UFH and bivalirudin reduced thrombus formation from DAPT alone by 66% and 76%, respectively (Fig. 6a, b).

Low-dose variegin ($7.7 \pm 2.6$ min) or ultravariegin ($6.2 \pm 1.3$ min) with DAPT did not increase the bleeding time compared to DAPT without anticoagulation ($6.4 \pm 3.7$ min). In contrast, UFH or bivalirudin with DAPT both resulted in the maximally observed bleeding time of 45 min. Figure 6c shows that, in the presence of DAPT, UFH or bivalirudin resulted in at least fivefold higher bleeding times than either variegin or ultravariegin. Taken together, on a background of DAPT, low-dose variegin or ultravariegin have marginally greater efficacy in preventing stent thrombosis than UFH or bivalirudin but with substantially less bleeding.

**Reversal agents for ultravariegin were identified from a naive human antibody library.** The availability of a reversal agent would provide an additional safeguard against bleeding complications from anticoagulation. We, therefore, explored the potential of generating 'active and specific' reversal agents for ultravariegin. Screening against a naive human antibody library, seven antibodies were found to bind to biotinylated ultravariegin. These clones were expressed and purified as IgG antibodies. These antibodies dose-dependently reversed ultravariegin's inhibition of thrombin's amidolytic activity (Fig. 7a). Almost complete reversal of 0.5 nM ultravariegin occurred with two antibodies (Ab1282 and Ab1283). The binding affinity between biotinylated ultravariegin and Ab1282 or Ab1283, measured by biolayer interferometry (BLI), was 1.25 and 1.40 nM (mean of two independent experiments), respectively (Fig. 7b, c and Supplementary Fig. 18). In contrast, Ab1283 did not show binding to the control peptide with a scrambled non-functional ultravariegin sequence, suggesting binding of Ab1283 to ultravariegin is sequence-specific (Supplementary Fig. 19). Next, the reversal activity of Ab1283 was tested in vivo. Rats that were injected with 0.75 mg/kg ultravariegin and followed by 10 mg of Ab1283 showed a statistically significant reduction in tail bleeding time (12.7 min vs 16.8 min, $P = 0.0078$) compared to those receiving saline in place of Ab1283 (Fig. 7d).

## Discussion
Despite the availability of direct oral anticoagulants (DOACs), anticoagulation for the acute coronary syndrome (ACS) and PCI still require parenteral anticoagulants that are effective and safe[7]. Concerns about an unfavourable efficacy-safety profile had led to reduced interest in thrombin as an anticoagulant target[6]. Our study revisits thrombin as a therapeutic target and identifies drug affinity-dose relationships as a means of optimising the therapeutic window of DTIs. Using complementary approaches including activity-based purification[29], cDNA amplification and data mining of tick saliva transcriptomes[30], we identified multiple potent and specific inhibitors of thrombin from *A. variegatum*. Starting with variegin, we progressively improved and developed ultravariegin, a 30-amino acid peptide with $K_i = 4.0$ pM with at least 1,000,000-fold selectivity for thrombin over other serine proteases. Compared with UFH and bivalirudin, variegin, and particularly ultravariegin, showed more gradual inhibition of thrombin generation across a broad range of drug concentrations in human blood, suggesting a capacity to regenerate thrombin for haemostasis when needed. In a series of efficacy and safety experiments in small and large animals, comparing variegin and ultravariegin with UFH and bivalirudin, we established that

variegin and ultravariegin had better antithrombotic efficacy with a shorter bleeding time when used at low doses in combination with DAPT. Importantly, variegin and ultravariegin achieved this therapeutic advantage with a single bolus injection at a fraction of the molecular weight-adjusted dose of a bivalirudin infusion. Finally, two specific reversal agents showing nanomolar level binding affinity to ultravariegin were identified. In vivo reversal experiments of rats anticoagulated with ultravariegin demonstrated that Ab1283 reduced bleeding time compared with control, representing a potential lead for antidote development.

Our results have several implications for peri-PCI anticoagulant drug development. First, we confirm that potent DAPT (aspirin and ticagrelor) without anticoagulation cannot sufficiently protect against stent thrombosis in an ex vivo porcine model. Second, thrombin remains a viable target for peri-PCI anticoagulation in the presence of DAPT. Here we showed that short, peptidic DTIs could be structurally optimised to inhibit thrombin with $K_i$ values 6- (variegin) and 445-fold (ultravariegin) better than bivalirudin, respectively. The strong affinity and specificity of variegin and ultravariegin allow for aggressive reduction of DTI dose in the presence of potent DAPT.

For any given therapeutic, high selectivity and affinity for the intended target are often crucial in achieving efficacy with minimum side effects[46]. In this regard, variegin and ultravariegin are substantially more advantageous than UFH and bivalirudin. Mechanistically, UFH is a heterogenous compound, indirectly inhibiting thrombin through antithrombin-III and is non-selective, resulting in unpredictable pharmacodynamics and pharmacokinetics[47]. These limitations are not present in specific DTIs including bivalirudin, variegin and ultravariegin. Previously, we have reported the novel class of DTIs from *A. variegatum* have some similarities to bivalirudin in structure but with several distinct features. Several structural–functional properties of variegin-like peptides may account for these differences from bivalirudin in pharmacokinetic-pharmacodynamic relationships. For example, variegin-like peptides contain specific amino acid residues within the active site and extended amino- and carboxyl-termini leading to stronger binding of thrombin. In contrast, bivalirudin has a non-specific, flexible tetra-glycyl linker between the active site and exosite-I binding sequence, resulting in loss of activity upon cleavage by thrombin[29–32]. Using a pair of peptides, UV011 and BV001, we have demonstrated continual, albeit weaker inhibition of thrombin after cleavage of ultravariegin, in contrast to complete loss of inhibition by bivalirudin after cleavage (Fig. 2d). In historical clinical trials, bivalirudin has previously been shown to increase the risk of myocardial infarction and stent thrombosis compared with UFH while the reduction in bleeding with bivalirudin might have been a result of unbalanced use of glycoprotein IIb/IIIa inhibitors with more patients in the UFH group receiving glycoprotein IIb/IIIa inhibitors than in the bivalirudin group[48]. More contemporary trials suggest that acute stent thrombosis may be mitigated with a prolonged infusion of full-dose bivalirudin post-PCI[12,14,49,50]. These more recent trials also had more balanced use of glycoprotein IIb/IIIa inhibitors across treatment groups and higher use of platelet P2Y$_{12}$ antagonists. With the exception of VALIDATE-SWEDEHEART, these trials showed that bivalirudin was associated with slightly less major bleeding compared with UFH with no difference in ischaemic events, indicating that thrombin remains an appropriate target for peri-PCI anticoagulation. The issues with bivalirudin appear to be related to efficacy and hence the dose during post-PCI infusion and concurrent use of antiplatelets are critical.

Considering the important role of platelets in arterial thrombosis, this approach of combining a low-dose, high-affinity anticoagulant with DAPT may achieve high antithrombotic efficacy while minimising bleeding risk during PCI. In the porcine

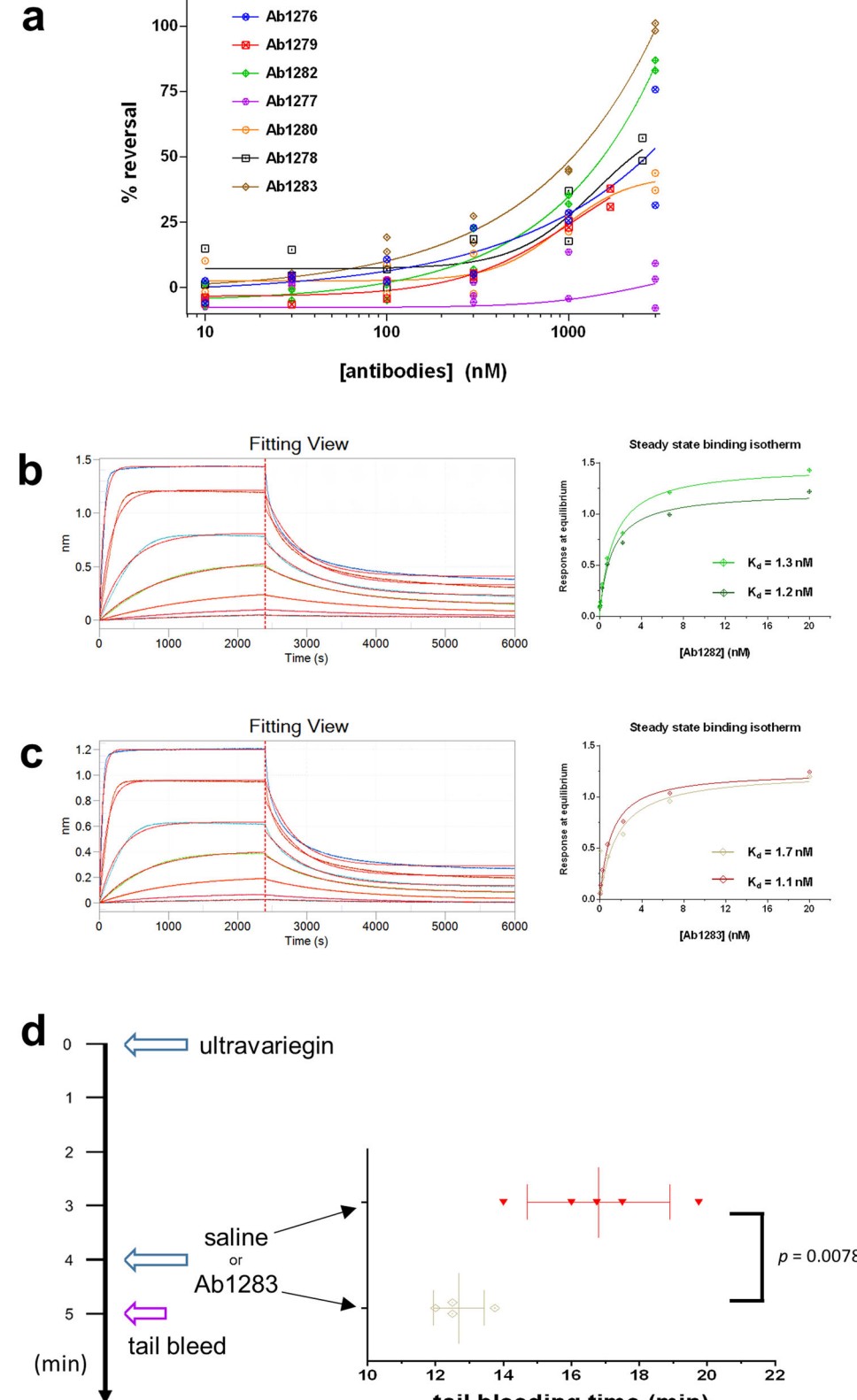

**Fig. 7 Reversal agents for ultravariegin. a** Antibodies discovered by screening against a human naïve antibody phage-display library dose-dependently reversed ultravariegin's (0.5 nM) inhibition of thrombin's (0.8 nM) amidolytic activity ($n = 2$ independent experiments for each antibodies, except Ab1277, for which $n = 3$). Representative BLI-binding sensorgrams between biotinylated ultravariegin and **b** Ab1282 or **c** Ab1283 fitted to a 1:1 kinetic model. Concentrations of Ab1282 and Ab1283 used were 20, 6.7, 2.2, 0.74, 0.25, 0.082 and 0.027 nM in a series. Steady-state analyses were performed to obtain dissociation constant $K_D$ at equilibrium ($n = 2$ independent BLI experiments). **d** Tail bleeding time from rats injected with 0.75 mg/kg ultravariegin at $t = 0$ min, and subsequently saline (red, $n = 5$) or 10 mg Ab1283 (beige, $n = 4$) at $t = 4$ min. $P = 0.0078$ from two-tailed Student's $t$ test between the two groups. The incision for tail bleed starts at $t = 5$ min. Data shown are mean ± SD.

model, UFH plasma levels corresponded to ~0.6 μM based on clotting time assays (Supplementary Figs. 20 and 21). The plasma concentration of bivalirudin was reported to be around 2.7 μM for PCI[51]. For 0.1 mg/kg variegin used in combination with DAPT, the average plasma concentration was ~30 nM[33]. Since ultravariegin was given at a fourfold lower dose compared with variegin, the plasma concentration of ultravariegin was estimated to be 7.5 nM. At low doses, variegin and ultravariegin appeared more potent than UFH and bivalirudin (Fig. 3a) without impacting the rate of synthesis of various enzymes/complexes important to haemostasis, thus causing less bleeding (Fig. 3b). In contrast, 0.6 μM UFH and 2.7 μM bivalirudin resulted in severe impairment of coagulation (e.g., min1 was less than half of the control); a decrease in the absolute value for min1 is clinically associated with an increase bleeding risk (Fig. 3b)[36]. The TGT also showed that the anticoagulant intensity of UFH and biva-lirudin increased sharply over a very narrow concentration range (Fig. 3, Supplementary Fig. 16 and Supplementary Table 2), suggesting a tendency to excessively suppress haemostasis with just small increases in their plasma concentration. By compar-ison, variegin or ultravariegin produced a more moderated increase in anticoagulation response over a wider concentration range. In in vitro assays, cleavage of ultravariegin results in two species of thrombin inhibitors: ultravariegin with higher affinity ($K_i = 4.0$ pM) and UV011 with 415-fold weaker affinity ($K_i = 1660$ pM). Hypothetically, UV011 may 'compete' with ultra-variegin for thrombin binding and hence an overall moderated increase in anticoagulation response was observed. In contrast, bivalirudin's cleavage product does not bind to thrombin, and any increase or decrease in the concentration of bivalirudin is sharply reflected in its anticoagulation intensity (Fig. 3, Supple-mentary Fig. 16 and Supplementary Table 2). Furthermore, in our rodent models, variegin/ultravariegin also achieved a three- to sevenfold wider therapeutic index compared with UFH/ bivalirudin.

Our study has limitations. First, the experiments using human blood were performed in vitro. Although we used thrombin-generation parameters that are widely validated to predict bleeding in clinical settings[52], the in vitro findings may not fully reflect thrombin regeneration capacity in vivo. Second, we used established small and large animal models to test the efficacy and safety of the four compounds but the results will need subsequent verification in human trials. Pre-clinical animal bleeding models may not always predict clinical bleeding risks in human trials because of a variety of reasons[53,54]. Here, we used three different bleeding models (tail, saphenous vein and superficial ear vein) to increase confidence in the results. Third, we did not perform dose optimisation of UFH and bivalirudin when used together with DAPT, since we aimed to validate variegin/ultravariegin in comparisons with clinically approved doses of UFH and bivalir-udin. It remains possible that dose optimisation will result in a more balanced efficacy-to-bleeding profile of UFH or bivalirudin. Fourth, although the two antibodies were able to fully reverse the thrombin inhibition effect of ultravariegin in vitro, the in vivo studies showed that large doses of Ab1283 may be required for complete reversal, hence the molecule may require further opti-misation as an antidote.

In conclusion, we discovered a unique class of bivalent, non-covalent DTIs from tick saliva transcriptomes and showed that optimisation of drug affinity-dose relationships can vastly improve the efficacy-safety balance of DTIs in ACS and PCI. Compared with UFH and bivalirudin, this new class of DTIs achieved greater efficacy at preventing ex vivo stent thrombosis but caused far less bleeding. We also demonstrated proof-of-concept in antidote development by identifying two antibody antidotes that effectively reversed the effect of ultravariegin

in vitro. These results require corroboration in clinical trials to determine if the salutary effects of this relatively simple affinity-dose optimisation approach is translatable to humans.

## Methods

**Study approvals.** All animal experiments were approved and conducted in accordance with the guidelines of the National University of Singapore (NUS) Institutional Animal Care and Use Committee (R16-008 and R15-0165). All human studies were approved by the NUS Institutional Review Board (B-15-094) and in compliance with the declaration of Helsinki. All participants gave written informed consent prior to study participation.

**Peptide synthesis and purification.** Peptides were synthesised using an auto-mated microwave peptide synthesiser (CEM, NC, USA). The C-terminal amino acid was loaded to Cl-MPA ProTide resin (CEM, NC, USA). Subsequent coupling of amino acids (0.2 M) were performed using 0.5 M N,N'-diisopropylcarbodiimide as activator and 0.1 M N,N'-diisopropylethylamine in 2 M Oxyma as activator base. Fmoc deprotection was achieved with 10% w/v piperazine in ethanol:N-methyl-2-pyrrolidone (1:9). Cleavage of the synthesised peptides were performed at room temperature for 3 h in the following cocktail: trifluoroacetic acid (TFA)/triisopro-pylsilane/water/dioxa-1,8-octane-dithiol (92.5:2.5:2.5:2.5). Peptides were pre-cipitated using cold diethyl ether. Purification of peptides was performed on Jupiter® 4 μm Proteo 90 Å (250 × 21.2 mm) reversed-phase column (Phenomenex, CA, USA). The purity and masses of peptides were determined on Aeris™ 3.6 μm Widepore XB-C18 100 Å (150 × 4.6 mm) reversed-phase column (Phenomenex, CA, USA) and by electrospray ionisation mass spectrometry (ESI-MS) using an LCQ Fleet Ion Trap MS and Thermo Xcalibur 2.2 software (Thermo Fisher Sci-entific, MA, USA) (Supplementary Figs. 1–14 and Supplementary Table 1). Mass spectra were deconvoluted using ProMass for Xcalibur 3.0 software. Concentra-tions of peptide solutions were estimated using UV absorbance at 280 nm and the extinction coefficient was calculated from the peptide sequence. For the two pep-tides without Tyr (avathrin and UV004), we measured absorbance at 205 nm and estimated the concentrations based on standard curves constructed using peptides of identical length and similar sequences with known concentrations (e.g., ultra-variegin and UV003). Ultravariegin and a peptide with scrambled ultravariegin sequence were synthesised with an additional Cys at the N-terminal for conjuga-tion to biotin using EZ-link™ iodoacetyl-LC-biotin (Thermo Fisher Scientific, MA, USA) using standard protocols recommended by the manufacturer.

**Inhibition of thrombin amidolytic activity.** Assays for the inhibition of thrombin (Haematologic technologies, VT, USA) amidolytic activity on chromogenic sub-strate S2238 (Chromogenix, NY, USA) by peptides were used to estimate half-maximal inhibitory concentration ($IC_{50}$) and inhibitory constant ($K_i$). All assays were performed in 50 mM Tris buffer (pH 7.4), 100 mM NaCl, and 1 mg/ml BSA at room temperature using 0.83 nM thrombin, varying concentrations of peptides and 100 μM S2238. The rates of formation of product p-nitroaniline were followed at 405 nm for 10 min with a InfinitePro M200 microplate reader using Tecan Magellan 7.0 software (Tecan, Mannedorf, Switzerland). Dose–response curves were fitted using Prism 6.0 software (GraphPad, CA, USA) to calculate $IC_{50}$ values with a logistic sigmoidal equation:

$$y = A_2 + (A_1 - A_2)/(1 + [x/x_0]^H) \qquad (1)$$

where $y$ is the percentage of inhibition, $A_2$ and $A_1$ are the right and left horizontal asymptote, respectively, $x$ is $\log_{10}$ of the inhibitor concentration, $x_0$ is the point of inflection and $H$ is the slope of the curve.

When an enzyme was inhibited by an equimolar concentration of inhibitor, the binding of the inhibitor to enzyme caused a significant depletion in the concentration of free inhibitors. To determine the apparent inhibitory constant, $K'_i$, the following tight-binding equation was considered:

$$v_s = (v_o/2E_t)(\{[K'_i + I_t - E_t]^2 + 4K'_i E_t\}^{1/2} - [K'_i + I_t - E_t]) \qquad (2)$$

where $v_s$ is the steady-state velocity in the presence of inhibitor, $v_o$ is the velocity observed in the absence of inhibitor, $E_t$ is the total enzyme concentration, $I_t$ is the total inhibitor concentration and $K'_i$ is the apparent inhibitory constant.

For competitive inhibition, the inhibitory constant, $K_i$, is related to $K'_i$ by this equation:

$$K'_i = K_i(1 + S/K_m) \qquad (3)$$

where $K'_i$ increases linearly with $S$, $K_i$ is the inhibitory constant, $S$ is the concentration of substrate and $K_m$ is the Michaelis–Menten constant for S2238.
For non-competitive inhibition, the inhibitory constant, $K_i$, is equal to $IC_{50}$.

**Serine protease specificity.** The selectivity profile of ultravariegin was examined against 13 serine proteases: fibrinolytic serine proteases (plasmin, TPA and uro-kinase), anticoagulant serine protease-activated protein C (APC), procoagulant serine proteases (FXIIa, FXIa, FXa, FIXa, FVIIa, kallikrein and thrombin) and classical serine proteases (chymotrypsin and trypsin). Effects of ultravariegin on these serine proteases were determined by inhibition of their amidolytic activities

assayed using chromogenic substrates specific for the respective enzymes. The final concentrations of proteases and substrates used are given in parentheses in nM and µM, respectively, unless mentioned otherwise: α-thrombin/S2238 (0.81/100), trypsin/S2222 (0.87/100), fIXa/Spectrozyme® fIXa (333/0.4), fXIa/S2366 (0.125/1000), fXa/S2765 (0.24/650), chymotrypsin/S2586 (1.2/0.67), tPA/S2288 (36.9/1000), fVIIa/S2288 (460/1200), plasmin/S2251 (3.61/1200), APC/S2366 (2.74/600), kallikrein/S2302 (0.93/1100), urokinase/S2444 (32 U/ml/650), fXIIa/S2302 (20/1000). The activity of thrombin was tested at lower concentrations of ultravariegin (10 nM, 1 nM and 0.1 nM) compared with other proteases (100 µM, 10 µM and 1 µM).

**Estimation for the molar concentration of UFH.** Three vials of 5000 unit/ml UFH from Leo Pharma (Ballerup, Denmark) were individually lyophilised to measure the average dry weight of UFH after subtracting stated additives. The average dry weight of UFH obtained after subtraction of the weight of excipients, was 136.3 mg, which is equal to 183.5 unit/mg UFH, consistent with reported values in the literature[55]. Assuming the average molecular weight of 10900 g/mol as reported[56], the equivalent molar concentrations of UFH were converted from U/ml in order to compare UFH with variegin, ultravariegin and bivalirudin, where data were plotted on the same axis.

**Plasma preparations.** Blood was collected by venipuncture from three healthy male donors into 3.2% trisodium citrate tube. Platelet-poor plasma (PPP) was obtained by centrifugation at 2000 × g for 10 min. Platelet-rich plasma (PRP) was obtained by centrifugation of the blood within an hour of sample collection at 180 × g for 10 min. The platelet count of PRP was adjusted to 150,000–200,000 platelets/µL with autologous PPP.

**Activated partial thromboplastin time (APTT) and clot waveform analysis (CWA).** APTT was performed on PPP using the Sysmex CS-5100™ (Kobe, Japan) using Actin FSL APTT reagent (Siemens Healthcare Diagnostics, IL, USA). PPP were spiked with variegin, ultravariegin, UFH, or bivalirudin for APTT according to the standard conditions recommended by the manufacturer. The final concentrations of aspirin and ticagrelor were 333 µg/ml and 3.33 µg/ml, respectively. Changes in light transmittance were recorded to determine the clot waveform[57]. Clotting times were as reported and the clot waveform was differentiated to derive the first, second and third derivatives (min1, min2 and min3, respectively)[38,57] using CWA analysis software IPU version 18.

**Thrombin-generation test (TGT).** Thrombin generation was determined using the Calibrated Automated Thrombogram (Diagnostica Stago, Asnières-sur-Seine, France) method in accordance with the manufacturer's instructions. In all, 980 µl of PPP or PRP were spiked with 10 µl of variegin, ultravariegin, UFH, or bivalirudin, and 10 µl of either DAPT (final concentrations 333 µg/mL aspirin and 3.33 µg/mL ticagrelor) or vehicle (5% dimethyl sulfoxide) accordingly. Preliminary dose titrations of aspirin and ticagrelor for inhibition of platelet aggregation in whole blood were performed using the Multiplate® Analyzer (Roche Diagnostics, Basel, USA). The lowest concentrations of aspirin and ticagrelor to produce more than 90% inhibition of platelet aggregation were selected for TGT. Thrombin generation of PRP and PPP was triggered using 1 pM and 5 pM tissue factor, respectively. Thrombin-generation curves were calculated using the Thrombinoscope 5.0 software (Thrombinoscope, Limburg, Netherlands).

Change in various thrombin-generation parameters with increasing doses differed among the four anticoagulants. The dose–response was plotted with double logarithmic axes to demonstrate linear relationships in anticoagulant intensity (represented by various TGT parameters) with changes in dose. Data were fitted by regression in Prism 6.0 according to the following equation:

$$Y = 10^{\wedge}[m^{*} \log(X) + C] \qquad (4)$$

where $Y$ is either LT, TTpeak, peak, ETP or VI, $X$ is the concentrations of anticoagulants, m is the slope and $C$ is the $Y$-intercept. The slopes calculated from curve fittings are recorded in Supplementary Table 2.

**Rats.** Male Sprague-Dawley rats (InVivos, Singapore) were housed in microisolator cages and were kept on a 12-h light/dark cycle with constant temperature and humidity. Rats were anaesthetised with a mixture of ketamine:xylazine (75:10 mg/kg body weight) and maintained with inhaled 1% isoflurane in oxygen throughout the experiment. Catheters (23 G) were inserted into the left femoral vein and artery, for drug injection and blood collection, respectively.

**Rat carotid artery thrombosis model.** Rat carotid artery thrombosis model[58] was performed as followed: five minutes after an i.v. bolus injection (saline, variegin, ultravariegin, or UFH) or 15 min after initiation of continuous i.v. infusion (bivalirudin), a 2 × 5 mm piece of filter paper (soaked in 4 µl 50% FeCl₃ solution) was placed on the surface of the carotid artery for 10 min. Blood flow through the common carotid artery was monitored with a Doppler flow probe (Model MA1PRB, Transonic System Inc., NY, USA) and recorded using LabChart 7 Pro (ADInstruments, CO, USA). Occlusion time was defined as the time taken after

FeCl₃ application for the blood flow to reach zero. The experiment was terminated after 60 min. Occlusion time was recorded as the maximal time of 60 min if no occlusion occurred by this time.

**Rat tail incision bleeding model.** Bleeding time[59] was measured in anaesthetised rats as followed: five minutes after an i.v. bolus injection (saline, variegin, ultra-variegin, or UFH) or 15 min after initiation of a continuous i.v. infusion (bivalirudin), a spring-loaded blade device (Surgicutt Adult bleeding time device, ITC, USA) was applied longitudinally on the ventral surface of the tail to make an incision (1 mm depth × 5 mm length) at 9–9.5 cm from the tip of the tail. The filter paper was used to blot blood from the side of the wound (without touching the wound) every 15 s. Bleeding time was defined as the time after incision until the cessation of bleeding on eight consecutive blots. The experiment was terminated at 60 min after tail incision. Bleeding time was recorded as 60 min if bleeding did not stop by this time.

This model was also used for testing the in vivo reversal activity of Ab1283. Four minutes after rats were injected with an i.v. bolus of 0.75 mg/kg ultravariegin, an i.v. bolus of 10 mg of Ab1283 or saline was injected. One minute later, an incision was made on the tail and bleeding time was measured as described above.

**Rat saphenous vein bleeding model.** The saphenous vein bleeding model[40,60] was performed in rats as followed: the right saphenous vein was exposed and covered with normal saline to prevent drying. Five minutes after an i.v. bolus injection (saline, variegin, ultravariegin, or UFH) or 15 min after initiation of continuous i.v. infusion (bivalirudin), a 23-G needle was used to pierce a hole in the right saphenous vein followed by a longitudinal incision of ~1 mm. Blood was gently wicked away every 15 s until haemostasis occurred. The clot formed was then gently removed using a 30-G needle to restart bleeding. The process of clot disruption was repeated after every incidence of haemostasis. The number of hae-mostasis events (i.e., clot formation) after repeated clot disruptions within a 30 min period were recorded. The average time per bleeding event were calculated by dividing 30 min with the number of clots (i.e., bleeding time).

**Pigs.** Both male and female SPF pigs (40–70 kg) were locally farmed for research purpose and obtained from the SEMC (SingHealth Experimental Medicine Centre, Singapore). All pigs were initially sedated with a mixture of ketamine (12 mg/kg), midazolam (0.5 mg/kg) and atropine (44 µg/kg). Endotracheal intubation was performed and anaesthesia was initiated with 5% isofluorane for three min, followed by 2% isofluorane for the duration of the experiment. Animals were ventilated using an Aespire ventilator machine (GE Healthcare, UK). Plasma replacement (6% volume) was given as a continuous infusion for the duration of the experiment. For the establishment of the extracorporeal loop, the carotid artery and jugular vein were surgically isolated and cannulated with 8 and 9 French percutaneous sheath introducers, respectively. Saline, variegin, ultravariegin, UFH, or bivalirudin, was administered through the cephalic vein cannula and blood were collected from the femoral artery. Real-time monitoring of ECG, heart rate, body temperature, respiration rate, mean arterial blood pressure, blood oxygen saturation and carbon dioxide levels was performed on a LifeWindow Lite multiparameter physiologic monitor (Digicare Biomedical Technology Inc., FL, USA).

**Preparation of the ex vivo perfusion chamber.** The descending aorta was removed from each terminated pig. The fascia and connective tissue were removed from the tunica externa. The aorta was cut into rectangular strips of 7 mm width × 27 mm length. Sharp forceps were then used to peel the smooth endothelial layer away from the rest of the vessel, exposing the tunica media. The thickness of the aortic strip was maintained at 0.5–0.6 mm, as measured using a calliper.

Perfusion chambers and stainless-steel connectors were custom manufactured (Sunway Precision Engineering, Singapore, Singapore). The perfusion chamber consists of two Plexiglas blocks, a bottom block (9 × 16 × 50 mm) with a 2-mm diameter tubular channel, and a top block (5 × 16 × 50 mm). A 28-mm section of the tubular channel was exposed in the middle section of the bottom block. A cobalt-chromium coronary stent (2.5 mm diameter × 23 mm length) was expanded to an outer diameter of 2 mm, pre-weighed and placed inside the tubular channel. A 7 × 27 mm strip of endothelium-denuded porcine aorta was placed on top to cover the exposed surface of the coronary stent. This simulated the contact area between the coronary stent and the coronary artery during PCI. The top block was then used as a cover for the bottom block and secured with a clamp. To connect the perfusion chambers to the tubing used in the extracorporeal loop, stainless-steel connectors (with a 2-mm diameter tubular channel) were screwed into the bottom Plexiglas block, directly in line with the tubular channel of the Plexiglas perfusion chamber.

**Porcine ex vivo stent thrombosis model.** For UFH, 100 U/kg was injected initially and 30 U/kg UFH top-up was used if needed to maintain the activated clotting time between 250 s and 350 s as per clinical practice guidelines[61]. Biva-lirudin was administered through an i.v. bolus injection of 0.75 mg/kg followed by a continuous infusion of 1.75 mg/kg/h. For variegin, either 1 mg/kg (without DAPT) or 0.1 mg/kg (with DAPT) was injected as a single bolus without additional doses. For ultravariegin, either 0.25 mg/kg (without DAPT) or 0.025 mg/kg (with

DAPT) were injected as a single bolus. DAPT (300 mg aspirin and 180 mg ticagrelor) was administered orally 16 h prior to surgery. The extracorporeal loop[41,62] was set up as followed: platinum-cured silicone tubing with 3.18 mm internal diameter (ID) × 6.35 mm outer diameter (OD) was connected to the carotid arterial (in) and jugular venous (out) cannulas (Supplementary Fig. 17). The loop was filled with saline prior to the experiment. Two minutes following drug or vehicle administration, blood was perfused through the extracorporeal loop with three consecutive perfusion chambers for 60 min at a flow rate of 70 ml/min driven by a peristaltic pump. After 60 min, the peristaltic pump was stopped and saline was perfused through the extracorporeal loop at a flow rate of 10 ml/min for 2 min to wash out residual blood and unbound debris. The perfusion chambers were disassembled, the aortic strips and stents were removed and their pictures were taken. The stents are washed with saline and gently dried using wipes. Stent thrombus was measured by weighing the stents with attached thrombus and subtracting the dry weight of the stent before the experiments. Using a scalpel blade, aortic thrombus was gently removed, washed with saline and dried with wipes, and then weighed again. Total thrombus weight was calculated as the sum of weights of the stent and aortic thrombi.

**Pig ear vein bleeding test**. The ear vein bleeding test was performed 5 min after administration of saline or anticoagulant[41]. An 18-G needle was used to puncture the ear vein and blood was dabbed from the wound every 20 s with a sterile gauze swab until bleeding has stopped. Dabbing of the wound was continued for 3 min after the last time-point that bleeding has stopped (no blood observed on gauze). The time between injury and cessation of bleeding was noted as the bleeding time.

**Screening of ultravariegin binders against phage-displayed naive human antibody library**. Biotinylated ultravariegin was screened against CIIDRET-PDHAL1, a naive human antibody library (10 billion clones) in the single-chain variable fragment (scFv) format[63,64]. In the first round of panning, ~$3 \times 10^{13}$ phages (3 ml in 2% BSA-PBS) were pre-adsorbed against streptavidin-coated MyOneT1 magnetic dynabeads. Biotinylated ultravariegin was added to a final concentration of 100 nM, followed by incubation for 2 h on a rotator at 5 rpm at room temperature. After binding, 100 μl streptavidin-coated M280 beads were added per ml of panning mixture (total 300 μl beads per 3 ml), followed by incubation for 30 min on a rotator at 5 rpm at room temperature. The beads were washed ten times each with PBST (PBS containing 0.05% Tween 20) and PBS and the bound phages were eluted using 100 mM Triethylamine, followed by neutralisation using 1 M Tris-HCl, pH 7.5. An aliquot of bound phages was titrated in *E. coli* TOP10F' cells, and the remaining phages were subjected to infection in TOP10F' cells. For the next round of panning, phages were rescued using AGM13 in 1200 ml volume and purified using double PEG precipitation. The second and third rounds of panning were performed as described above with appropriate modifications; the amount of antigen was reduced to 50 nM and 10 nM in the second and third round of panning, respectively, and the number of washes was increased to 15 each with a round of PBST and PBS washing. After the third round of panning, individual phage clones were analysed using phage ELISA and DNA sequencing and seven phage clones specific to biotinylated ultravariegin were identified.

**Cloning, expression and purification of antibodies against biotinylated ultravariegin**. Variable light and heavy chain genes encoded by selected anti-ultravariegin scFv(s) were sub-cloned in pcDNA3.4 based pVCLC102 (Kappa Light Chain; KLC) or pVCLC202 (Lambda Light Chain; LLC) and pVCHC302 (Heavy Chain; HC) vectors for expression of antibodies in IgG format. Purified LC and HC plasmid DNA were mixed 1:1 and transfected in 50/100 ml ExpiCHO cells using the Max Titer expression protocol as per manufacturer's instructions (Thermo Fisher Scientific, MA, USA). Culture supernatant was harvested 8 days after transfection, purified using 1 ml HiTrap MabSelect SuRe columns (GE Healthcare Life Sciences, IL, USA) and eluted with a linear gradient comprising 0.1 M citrate, pH 3.0 followed by neutralisation of pH using 2 M Tris-HCl. Concentrations of each protein were estimated by absorbance and extinction coefficient values at 280 nm, assuming molar extinction coefficient of human IgG to be 210,000 $M^{-1}$ $cm^{-1}$.

**Binding affinity measurements by biolayer interferometry**. The binding affinities of biotinylated ultravariegin and scrambled ultravariegin for the two reversal agents, Ab1282 and Ab1283, were measured by biolayer interferometry on an Octet RED96 System (Pall FortéBio, CA, USA) and data analysis performed using FortéBio Data Analysis 9.0 software. Biotinylated peptides were loaded onto Streptavidin (SA) biosensors. The experiments were carried out in 50 mM Tris buffer (pH 7.4) containing 100 mM NaCl and 1 mg/ml BSA at 27 °C. Typically, the biosensors were pre-equilibrated in the buffer for 750 s, loaded with biotinylated peptide for 900 s, re-equilibrated to baseline for 300 s before the association phase of 2400 s in wells containing 20 nM, 6.7 nM, 2.2 nM, 0.74 nM, 0.25 nM, 0.082 nM and 0.027 nM of Ab1282 or Ab1283. Reference wells contained no antibodies. Biosensors were transferred into new wells containing assay buffer for a dissociation phase lasting 3600 s. Binding kinetics were calculated using the FortéBio Data Analysis 9.0 software. The association ($k_{on}$) and dissociation ($k_{off}$) rate constants were obtained by fitting the association and dissociation data to a 1:1 model. The equilibrium dissociation constant, $K_D$, was estimated by fitting the steady-state binding isotherm of response at equilibrium ($R_{eq}$) against the concentrations of antibodies.

**Statistical analysis**. All statistical analyses and curve-fitting by non-linear regression were performed on Prism 6.0 (GraphPad, CA, USA). Dose–response fits for Table 1 and Fig. 7a, tight-binding fits (Morrison equation) for Fig. 2b–d and the log-log line fit for Supplementary Table 2 were used ascertained with goodness-of-fit analyses as implemented in Prism 6.0. The Pearson correlation coefficient, $r$, was used to calculate the correlation between bleeding times obtained in the tail bleeding and saphenous vein bleeding models (Fig. 4f, insert). One-way analysis of variance (ANOVA) and post hoc Tukey's multiple comparisons tests were used to compare the quantitative end-points between treatment groups in Figs. 5 and 6. Two-tailed, unpaired $t$ test was performed for comparison between treatment (Ab1283) and control (saline) groups in Fig. 7d. Multiplicity-adjusted $P$ values for each comparison were reported.

**Reporting summary**. Further information on research design is available in the Nature Research Reporting Summary linked to this article.

## Data availability
Transcriptomic data utilised have been previously published[34] and not generated in this paper. The sequence data used to inform the design of ultravariegin as depicted in Fig. 1 is publicly available in UniProt under accession number P85800 and in GenBank under accession number BAD29729.1, DAA34688.1, DAA34258.1 and DAA34160.1. Sequences of antibodies tested as reversal agents are available upon reasonable request under cover of a non-disclosure agreement until the application for intellectual property is completed and published. The remaining data generated in this study are provided in the Article, Supplementary Information or Source Data file. Source data are provided with this paper.

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

## Acknowledgements

The authors would like to thank Raavi Panwar and Apurva Kulkarni for technical assistance. This research was supported by grants from Singapore National Medical Research Council (NMRC) Clinician Scientist—Individual Research Grant (CIR3nov009 & CIRG17may014) (M.Y.C.), and NMRC Open Fund—Young Investigator Research Grant (16may021) (C.Y.K.).

## Author contributions

C.Y.K. and N.S. designed the study, performed experiments, analysed the data and wrote the manuscript. C.Y.C.Y., J.K.I. and V.V. performed experiments and analysed data. A.W.L.L., W.C., F.S.A., E.J.E.L., G.C. and M.I.B.M. performed experiments. Y.L.C., E.S.Y., D.M.M., M.H., R.C.B., D.P.V.K., A.G., V.K.C. and A.M.R. designed the study. R.M.K. and M.Y.C. designed the study, analysed the data and wrote the manuscript. All authors reviewed and approved the final version of the manuscript.

## Competing interests

Part of the work related to the peptides as thrombin inhibitors was included in the patent application with C.Y.K., J.K.I. and R.M.K. among the inventors (WO2016204696A1 Novel thrombin inhibitors). Antibodies tested as reversal agents for ultravariegin were included in another patent application with C.Y.K., V.V., A.G., V.K.C., R.M.K. and M.Y.C. among the inventors (application in progress). The remaining authors declare no competing interests.
