## [Peer Review File · Nature Communications]

Efficacy and safety of next-generation tick
transcriptome-derived direct thrombin inhibitorsREVIEWER COMMENTS

Reviewer #1 (Remarks to the Author):

The paper characterizes a new class of direct thrombin inhibitors derived from the salivary gland of the tropical bone tick and demonstrates superior selectivity for thrombin with superior inhibition as compared to bivalirudin with lower bleeding times. This suggests wider therapeutic index for these agents. Multiple in vivo models are used including the rat carotid thrombosis model and well as what the authors term a stent thrombosis model with both showing superior anti thrombotic activity as compared to bivalirudin and heparin with lower bleeding times at therapeutic doses. The suggest here is that this discovery of new DTIs may help to reduce bleeding while also improves the anti-thrombotic effects of these agents all of which are used during PCI and in other areas of medicine.

The work is clinically relevant as all the agents tested (Bivalirudin and UFH) are in wide clinical use and UFH is ubiquitous during percutaneous procedures. While I accept the authors may have found a new DTI-some of the model used are not well explained and the necessary controls used to make these claims to not seem to be in place. As well, models are inappropriately characterized and more careful description of the experiments is required. At present this manuscript cannot fully support the authors claims. My major comments are below

- 1) The authors overemphasize the ills of UFH and Bivalirudin. UFH has been in clinical use since the early 1900s and so far it is by far the most used anti-coagulant. It may have some flaws but it works very well despite some limitations. I would tone down the introduction.
- 2) Please describe in further detail the references 14 and 15. How much shown here is overlapped with that paper? Several figures look very similar. Indeed figure 3 in that work (14) looks to be the same series of experiments perhaps repeated this time? Please be transparent about what is new and what you have shown previously.
- 3) The rat carotid thrombosis model, there is no mention of the number of animals used for each experiments (simply n=3 to 8?). How was occlusion measured in the model? Doppler US? Where is the control (not therapy)? Were flow rates closely monitored as well as coagulation factors and platelet inhibition. What does response time mean>. Time to occlusion? This data needs much clear explanation
- 4) Bleeding time is a rather old measure and clinically no longer used and partially dependent upon platelet activity. Is there a better measure? It would be nice to show an illustration of the model and the outcomes measured. Also showing some kind of picture would be nice.
- 5) The "stent thrombosis model is not well characterized and I don't agree with the terminology used. This is an AV Shunt model with stents inserted into the shunt. No description of how the study was done? Was it one stent per shunt? How many pigs. Was the same pig used for all experiments. If so was the platelet function and platelet count and coagulation measured to make sure these did not get depleted during the shunt runs. How was thrombi weight obtained? How long was the shunt run for each experiment. There are so many detailing missing it is hard to evaluate this experiment at all. What about the flow rates from shunt to shunt at the beginning and end?
- 6) Same for the DAPT shunt. The graphs show very small differences in the amount of thrombus making me wonder if any of this is clinically relevant. Again the same problems I raised in question 5 arise.
- 7) The use of the aortas also need to be better explained. were these sitting above the stents? Some diagram or model would be nice. Where is the data on the aortas given? I am not sure I understand the use of the aortas here.

Reviewer #2 (Remarks to the Author):

The manuscript reports on the characterization of a family of thrombin inhibitors from tick saliva in numerous *in vitro* and *in vivo* assays. Authors have successfully modified one of the variants to achieve higher affinity and anti-thrombotic effects *in vivo*, and they went an extra mile using rats and porcine models, which is remarkable. Finally, antibodies against ultravariegin were selected, as a strategy to revert anticoagulant effects of the inhibitor. Overall, this study represents a remarkable amount of work, with findings supporting further development of ultravariegin for potential clinical studies.

Suggestions/Critiques

Introduction

- Specify the most common therapy for PCI in numbers, particularly the % of patients being treated with bivalirudin, compared to UFH, and also for the occurrence of bleeding. Which is the preferred protocol by intervention cardiologists? This adds on understanding and context of the magnitude of bivalirudin's use nowadays. Few words for cost is also a plus if available. Also, the half-life of variegin is indicated – how does it compare to bivalirudin. A word about presence of anti-thrombin from other species such as mosquitoes may also be useful for the non-specialist in vector biology.

Methods:

- Which human study was approved (line 382)?
- Provide synthesis and refolding protocols (if applicable) in greater detail, including the specific mass for each peptide compared to theoretical mass, after MS. What was the yield?
- Is the calculation of peptide concentration reliable, with no Tryptophan in the sequences (one Tyr is noted)? This is critical for the implications of the results, and need to be verified.
- Rational for the CWA is required, not a universally used method.
- How many rats and pigs for each experiment/dose?
- Specify the source for pigs, i.e., were they in captivity, and if so where, no information is provided.
- Explain why ACT was used, not PT or aPTT.
- Bivalirudin recommended dose is 0.75 mg/Kg/hr in humans. How did the authors decide the dose of peptides to be used (0.025-0.25 mg/Kg)?

Results, Figures:

- Figure 2A – specify if the inhibitor (ultravariegin) behaves as a slow or fast tight type inhibitor.
- Figure 2C – the projection of the linear regression is negative when it intercepts the y axis (also indicate the Unit in the axis - pM? nM?), which can be used to calculate a K_i . How this reconcile with a K_i of 4 pM?
- Figure 2E – it is expected 100% inhibition of 0.83 nM thrombin with 1 nM of a bivalent tight inhibitor such as ultravariegin, and even more so with 10 nM. This is not the case. It might be that the concentration of the reactants is somewhat not accurate. Please verify.
- Figure 3C – the actual plot for thrombin generation assay is needed, at least for ultravariegin.
- Figure 4D – Bivalirudin expressed in mg/Kg/h in the x axis because of continuous infusion. This makes any comparison more difficult as the peptides were tested in mg/Kg, as a bolus infusion. Did the author try bolus infusion of bivalirudin at similar concentration as other peptides so it can be compared? Please elaborate and discuss. Specifics regarding pharmacokinetics if available.
- Figure 5 and 6 – it is difficult to really extrapolate the results with respect to bleeding vs antithrombotic activity with the concentrations tested, and imply safety of one molecule vs another. It could be that bivalirudin at 0.3 mg/Kg/hr shows the same antithrombotic effect at the tested concentration 0.75 mg/Kg, but would cause much less bleeding. Evidently, these experiments are exceptionally costly and time consuming for a dose-response curve, but few words in this respect are reassuring without diminishing the implications of authors' findings. How n in figure legends indicate number of animals per data point? Please clarify.

Discussion

- Lines 355 – 359 – please elaborate why bivalirudin has a narrow therapeutic window compared to variegin?
- Atherosclerotic plaque contain Tissue Factor, can the authors elaborate on strategies targeting TF in the therapy for PCI?

- Overall, this is an impressive work involving numerous areas of expertise that provides proof of concept to develop ultravariengin to the next step which is its potential clinical use. It also underlines the importance of the study of salivary components from blood-feeding arthropods as tools in biochemistry and in the development of novel therapeutics. Finally, please include your changes/replies to the reviewer, to the manuscript.

Reviewer #3 (Remarks to the Author):

As instructed, I have limited my peer-review to the ligands (peptides and antibodies) used. I have two concerns:

1. There is no QA/QC for the synthetic peptides. As such it is challenging to determine purity of the preparation and rule out potential artifacts.
2. While the authors appropriately "pre-empt" the criticism that the seven monoclonal antibodies selected from a naive phage library were only tested in vitro in the end of the manuscript Discussion, instead they should really test at least one of their leading candidates in vivo and demonstrate reversal of their DTIs.

Reviewer #4 (Remarks to the Author):

Koh and co-workers have characterized potential thrombin-inhibiting peptides present in tick saliva and then optimized amino acids from one of these peptides, variengin, to produce a final peptide called ultravariengin. Ultravariengin inhibits thrombin with a K_i of 4 pM but has little affinity for other coagulation proteases. The anticoagulant activity of variengin and ultravariengin were compared to UFH and bivalirudin in a series of coagulation assays performed in human plasma or human platelet rich plasma. Their ability to inhibit clotting and produce bleeding were examined using in vivo/ex vivo assays in rats and pigs. Ultravariengin reduced thrombin formation more gradually than UFH or bivalirudin in the plasma clotting assays. In animal experiments, ultravariengin prevented thrombosis while producing much less bleeding than UFH or bivalirudin (Figures 6B & C). The findings are of interest because these pre-clinical data suggest that these tick-derived peptides may have much better efficacy to safety profiles than previously developed direct thrombin inhibitors. Thus, these studies advance the field for further development of this class of anticoagulants for use in percutaneous coronary interventions. I have the following comments for the authors.

1. The thrombin generation data indicate that ultravariengin inhibits thrombin formation more gradually as its concentration is increased when compared to bivalirudin with the largest differences in the change in lag time and change in time to peak thrombin. It is concluded that this gradual inhibition preserves a capacity to regenerate thrombin for haemostasis when needed, and potentially explains why ultravariengin has better antithrombotic efficacy with less bleeding than UFH or bivalirudin. This conclusion deserves further explanation from the authors.
 - a. How does the mechanism inhibition of thrombin by ultravariengin differ from that of bivalirudin that accounts for the unique effects of ultravariengin?
 - b. Why does inhibition of thrombin generation by ultravariengin allow for regeneration of thrombin in a tail clip or ear puncture wound but not on a stent or de-endothelialized vasculature.
2. A concern with all bleeding models in animals is that they poorly predict actual bleeding in humans. The data are compelling that in the models studied ultravariengin produces less bleeding than UFH or bivalirudin. Nevertheless, additional discussion of the weaknesses of the two bleeding models used is warranted in the limitations paragraph in the discussion.

REVIEWER COMMENTS

Reviewer #1 (Remarks to the Author):

The paper characterizes a new class of direct thrombin inhibitors derived from the salivary gland of the tropical bone tick and demonstrates superior selectivity for thrombin with superior inhibition as compared to bivalirudin with lower bleeding times. This suggests wider therapeutic index for these agents. Multiple in vivo models are used including the rat carotid thrombosis model and well as what the authors term a stent thrombosis model with both showing superior anti thrombotic activity as compared to bivalirudin and heparin with lower bleeding times at therapeutic doses. The suggest here is that this discovery of new DTIs may help to reduce bleeding while also improves the anti-thrombotic effects of these agents all of which are used during PCI and in other areas of medicine.

The work is clinically relevant as all the agents tested (Bivalirudin and UFH) are in wide clinical use and UFH is ubiquitous during percutaneous procedures. While I accept the authors may have found a new DTI-some of the model used are not well explained and the necessary controls used to make these claims to not seem to be in place. As well, models are inappropriately characterized and more careful description of the experiments is required. At present this manuscript cannot fully support the authors claims. My major comments are below

We thank the Reviewer for the comments and have revised the manuscript accordingly. Line numbers used to refer to specific sections of the manuscript are based on the tracked changed version of the manuscript file.

1) The authors overemphasize the ills of UFH and Bivalirudin. UFH has been in clinical use since the early 1900s and so far it is by far the most used anti-coagulant. It may have some flaws but it works very well despite some limitations. I would tone down the introduction.

We agree that UFH remains an important anticoagulant in PCI despite several theoretical drawbacks. We have added new sentences in introduction: *"However, UFH remains one of the most widely used parenteral anticoagulants due to low cost (approximately \$4 to \$10 USD per PCI⁵) and the wealth of clinical experience accumulated in more than eight decades of use⁶. To overcome some of UFH's disadvantages, an injectable..."* (Page 4, line 84-87) to emphasise the major advantages of UFH. However, we also think the text in our introduction fairly reflects current clinical practice guidelines and consensus concerning bivalirudin, as well as the need for better antithrombotic strategies.

We have further modified the end of the first paragraph to explain the motivation for our study: *"With increasing use of potent platelet P2Y₁₂ antagonists such as ticagrelor, prasugrel, and cangrelor, there remains an even greater unmet is a need for safer peri-PCI anticoagulants to adequately improve the efficacy-safety balance of antithrombotic therapy during PCI^{7,17}. Combination antiplatelet and anticoagulant therapy has become more common especially among patients with atrial fibrillation or venous thromboembolism undergoing PCI^{15,18}."* (Page 5, line 102-107).

2) Please describe in further detail the references 14 and 15. How much shown here is overlapped with that paper? Several figures look very similar. Indeed figure 3 in that work (14) looks to be the same series of experiments perhaps repeated this time? Please be transparent about what is new and what you have shown previously.

Reference 14 and 15 (29 and 30 in the revised manuscript) describes the discovery and characterisation of varieggin and avathrin, respectively. As stated in the 2nd paragraph of the introduction (Page 5), these are peptidic DTIs we have previously identified in the salivary gland of the tropical bont tick. Varieggin was initially identified in the proteome of salivary gland extracts (previously

Ref 14) and avathrin was identified from cDNA amplification using salivary gland tissues (previously Ref 15).

Subsequently, different set of sequences were identified through searches in the database and one of the sequences representing a short repeat in the sequence identifier DAA34688.1 was synthesised. We studied structure-function relationships by progressively designing and synthesizing different peptides ultimately leading to ultravariagin, as described in Page 7 to 9 (line 139-198). Sequences of all these peptides, including bivalirudin, variagin and avathrin, are listed in Table 1. All experiments depicted in Fig. 2 are new experiments conducted on new peptides designed and synthesised within the scope of this manuscript (11 new peptides). In Table 1, the K_i calculated for the 3 previously published peptides (bivalirudin, variagin, and avathrin) are based on new experiments performed within the scope of this study and reported for consistent comparisons with the 11 new peptides. K_i values of the 3 peptides are all similar to previously reported values. These enzymatic inhibition assays are typical experiments reported for inhibitors of coagulation factors, especially thrombin inhibitors, and all methods are described in detail under the “**Inhibition of thrombin amidolytic activity**” (Page 22 & 23, line 494-521) and “**Serine protease specificity**” (Page 23 & 24, line 522-536) sections in **Materials and Methods**. Specifically, Fig 3 in Ref 14 (initial submission) is a screen of variagin against various coagulation serine proteases while Fig. 2e in the current manuscript was done using the new peptide ultravariagin in a similar setup. In this revision, we have also replaced the bar chart in Fig. 2e with a scatter plot.

3)The rat carotid thrombosis model, there is no mention of the number of animals used for each experiments (simply n=3 to 8?). How was occlusion measured in the model? Doppler US? Where is the control (not therapy)? Were flow rates closely monitored as well as coagulation factors and platelet inhibition. What does response time mean>. Time to occlusion? This data needs much clear explanation

We have now included the number of animals used for each data point in the legend of Fig. 4 (Page 34 & 35, Line 785-809). Details of the experiments are included under the “**Rats**”, and “**Rat carotid artery thrombosis model**” sections in **Materials and Methods** (Page 25 & 26, Line 581-596). Briefly, occlusion was measured a Doppler flow probe (Model MA1PRB, Transonic System Inc., USA). Occlusion time was defined as the time taken for the blood flow to reach and stabilised at zero. We did perform the experiments using saline as control; however, since Fig. 4a to 4d are a series of dose-response plots with x-axis on log-scale, we apologise for not being able to plot the data points for anticoagulants = 0. In the revised manuscript, we have used dashed lines to represent the baseline (control saline) values for both carotid artery thrombosis (red line) and bleeding (blue line) models. We did not monitor platelet inhibition in rat carotid artery thrombosis model, in line with typical reports for this model.

Fig 4a to 4d were plotted with data points from two separate animal models on each graph: (1) carotid artery thrombosis (observing time to occlusion) and (2) tail bleeding (observing bleeding time). The duration of observations (y-axis) for both models were standardised at 60 min, therefore, we used a single Y-axis to report the “response time” for both occlusion (red line) and bleeding (blue line). This facilitates the visualisation of the separation between efficacy (occlusion, red) and safety (bleeding, blue) at any given dose of anticoagulant on the corresponding x-axis. We have added these sentences into Fig. 4 legend (Page 34, line 787-789) to clarify the title of the y-axis. Detailed descriptions for thrombosis and bleeding models can be found under “**Rat carotid artery thrombosis model**” and “**Rat tail incision bleeding model**” in **Materials and Methods** (Page 26 & 27, line 587-606).

4)Bleeding time is a rather old measure and clinically no longer used and partially dependent upon platelet activity. Is there a better measure? It would be nice to show an illustration of the model and the outcomes measured. Also showing some kind of picture would be nice.

We agree with the reviewer that tail bleeding time may have certain limitations, especially in predicting the clinical setting in humans (eg. template bleeding time). However, in pre-clinical animal studies, tail bleeding time is the most commonly used assessment of bleeding risk. (Greene *et al. J Thromb Haemost* 2010, 2820-2822; Mohammed *et al. Platelets* 2020, 417-422). More importantly, tail bleeding time have been reported in pre-clinical animal studies of anticoagulants currently on the markets (eg. Morishima *et al. Thromb Res* 2013, 234-239, for edoxaban; Wiene *et al. Thromb Haemost* 2007, 333-338, for dabigatran). Therefore, we have elected to use the tail bleeding model as one of the bleeding models studied in the manuscript. To confirm the robustness of our results, we have also used an ear vein bleeding assay in pig as a complementary model to assess bleeding risk.

However, we acknowledge that additional bleeding models can strengthen our study. Therefore, we have now performed additional experiments using a third bleeding model (saphenous vein bleeding model) that have been demonstrated to be sensitive to bleeding disorders related to abnormalities in coagulation (eg. Ay *et al. J Thromb Haemost* 2017, 1829-1833; Monroe and Hoffman *Thromb Res* 2014, S6-S8). The primary endpoint for the saphenous vein bleeding model is the number of haemostatic events (ie. clot formation) following repeated destruction of clots. The higher the number of haemostatic events, the more preserved the haemostatic capacity. We now demonstrate that average time between haemostatic events (ie bleeding time) showed strong correlation with tail bleeding time. For the four anticoagulants (variegin, ultravariegin, UFH, and bivalirudin), at doses that resulted in 50% efficacy in the carotid artery thrombosis models, the rank ordering of bleeding risk are identical (ie. tail bleeding time ultravariegin < variegin < bivalirudin < UFH vs haemostatic events ultravariegin > variegin > bivalirudin > UFH). Results from the new bleeding model have now been added as Fig. 4e and 4f, and as text in the **Results** section (Page 12, Line 274-284). Detailed descriptions for the three bleeding models used can be found under "**Rat tail incision bleeding model**" (Page 26 & 27, line 597-610) and "**Rat saphenous vein bleeding model**" (Page 27, line 611-622), and "**Pig ear vein bleeding test**" (Page 30, line 680-686) in **Materials and Methods**. Unfortunately, we do not have the approval to take photographs of animals under our local IACUC regulations but similar pictures can be found in cited references.

5)The "stent thrombosis model is not well characterized and I don't agree with the terminology used. This is an AV Shunt model with stents inserted into the shunt. No description of how the study was done? Was it one stent per shunt? How many pigs. Was the same pig used for all experiments. If so was the platelet function and platelet count and coagulation measured to make sure these did not get depleted during the shunt runs. How was thrombi weight obtained? How long was the shunt run for each experiment. There are so many detailing missing it is hard to evaluate this experiment at all. What about the flow rates from shunt to shunt at the beginning and end?

The porcine model is an ex vivo model of thrombosis using a modified perfusion chamber, known as the Badimon perfusion chamber, with stents placed in the perfusion chambers. It is commonly used to investigate thrombosis when blood is exposed to thrombogenic tissue under flow conditions in animals or humans. Importantly, this model has been reported in multiple studies of antithrombotic drugs development in large animals and humans (eg. Becker *et al. J Thromb Haemost* 2012, 2470-2480, for rivaroxaban, aspirin, and clopidogrel; Vilahur *et al. Circulation* 2004, 1686-1693, for LA816, aspirin, and clopidogrel; Wilson *et al. Cardiovasc Res* 2019, 669-677, for JNJ-64179375). The name "porcine model of stent thrombosis" is consistent with the name used in Becker *et al.* but we have now modified it to "porcine ex vivo stent thrombosis model" for better clarity. Detailed procedures of the model has been described across several sections under **Materials and Methods** in the initial submission, including the preparation of pigs (under "**Pigs**", Page 27 & 28, line 623-637), the construction and preparation of the perfusion chambers (under "**Preparation of the ex vivo perfusion chamber**", Page 28 & 29, line 638-656), the procedures of extracorporeal perfusion experiments and weighing of stents and aorta thrombi (under "**Porcine ex vivo stent thrombosis model**", Page 29 & 30, line 657-679), and the ear vein bleeding model (under "**Pig ear vein bleeding test**", Page 30, line 680-686).

6) Same for the DAPT shunt. The graphs show very small differences in the amount of thrombus making me wonder if any of this is clinically relevant. Again the same problems I raised in question 5 arise.

The dose of DAPT, UFH and bivalirudin used are consistent with clinical practice guidelines for PCI, therefore it is not too surprising that maximum antithrombotic efficacy were achieved, resulting in the small differences in the amount of thrombus formed (Fig. 6b). When Fig. 5 & 6 are considered together, it showed that in the highly thrombogenic environment of PCI, simulated by endothelium-denuded pig aorta strip in the perfusion chambers, there is still room for improvement of efficacy of UFH and bivalirudin even at clinically relevant doses. In contrast, ultraparicillin achieved better efficacy (95% thrombus reduction) than UFH and bivalirudin ($p \leq 0.0001$ & $p \leq 0.05$, respectively, Fig. 5b). Maximum efficacy may be achieved by UFH and bivalirudin in combination with DAPT (Fig. 6b) but comes with a price of increased bleeding (Fig. 6c). With the high affinity of ultraparicillin for thrombin, the dose of ultraparicillin can be substantially lowered to reduce bleeding risk (Fig. 6c) while achieving the same efficacy as UFH and bivalirudin (Fig. 6b), when combined with DAPT.

Similar to our answers for comment 5, all procedures for the model are now described in detail under the **Materials and Methods** section in the resubmitted manuscript.

7) The use of the aortas also need to be better explained. were these sitting above the stents? Some diagram or model would be nice. Where is the data on the aortas given? I am not sure I understand the use of the aortas here.

Endothelium-denuded pig aorta exposed to blood flowing through the perfusion chambers with thrombogenic surfaces simulate vascular injury during stent placement. The aorta strips are indeed placed above the stents. The dimensions of the perfusion chambers, preparation of the aorta strips, stent and aorta placement, as well as weighing of stent and aorta thrombi were all described in detail under the **Materials and Methods** section in the resubmitted manuscript (Page 27 to 30, line 623-686). A figure depicting the model is included as Supplementary figure 17.

Reviewer #2 (Remarks to the Author):

The manuscript reports on the characterization of a family of thrombin inhibitors from tick saliva in numerous in vitro and in vivo assays. Authors have successfully modified one of the variants to achieve higher affinity and anti-thrombotic effects in vivo, and they went an extra mile using rats and porcine models, which is remarkable. Finally, antibodies against ultravariegin were selected, as a strategy to revert anticoagulant effects of the inhibitor. Overall, this study represents a remarkable amount of work, with findings supporting further development of ultravariegin for potential clinical studies.

We thank the reviewer for the kind encouragement. In our response, line numbers are based on the tracked change manuscript file.

Suggestions/Critiques

Introduction

- Specify the most common therapy for PCI in numbers, particularly the % of patients being treated with bivalirudin, compared to UFH, and also for the occurrence of bleeding. Which is the preferred protocol by intervention cardiologists? This adds on understanding and context of the magnitude of bivalirudin's use nowadays. Few words for cost is also a plus if available. Also, the half-life of variegin is indicated – how does it compare to bivalirudin. A word about presence of anti-thrombin from other species such as mosquitoes may also be useful for the non-specialist in vector biology.

We have added all the requested information in the **Introduction**, including the addition and modifications to the following sentences:

Page 4, line 84-86: *“However, UFH remains one of the most widely used parenteral anticoagulants due to low cost (approximately \$4 to \$10 USD per PCI⁵) and the wealth of clinical experience accumulated in more than eight decades of use⁶.”*

Page 4, line 88-92: *“Although bivalirudin is more expensive (approximately \$400 to \$600 per PCI without post-procedural infusion⁵), initial randomized trials showed that bivalirudin was associated with similar antithrombotic efficacy but less bleeding when compared with a combination of UFH and a platelet glycoprotein IIb/IIIa inhibitor^{8,9,10}.”*

Page 4 & 5, line 94-102: *“More recent trials with balanced use of glycoprotein IIb/IIIa inhibitors in both bivalirudin and UFH arms have shown less favourable results for bivalirudin^{11,12,13,14}. Therefore, in the 2018 European Society of Cardiology/European Association for Cardio-Thoracic Surgery guidelines on myocardial revascularization, routine use of UFH received a higher recommendation (class I) than bivalirudin (class IIb) for peri-PCI anticoagulation¹⁵. Bivalirudin has been more widely adopted in the United States than the rest of the world. One study estimated that 47.1% of PCI cases between July 2009 and December 2014 in the US used bivalirudin (52.9% UFH) but also noted a decline in bivalirudin usage after 2013¹⁶.”*

Page 5, line 108-112: *“Haematophagous animals such as leeches, mosquitoes, ticks, tsetse flies and others are rich source of antithrombotic agents¹⁹. Molecules such as hirudin from the medicinal leech²⁰, anophelin from mosquitoes^{21,22}, madanin from ticks²³, and tsetse thrombin inhibitor from tsetse flies^{24,25} are potent and selective thrombin inhibitors that are highly amenable to customisation as synthetic inhibitors with improved properties^{25,26,27,28}.”*

Page 5, line 117-121: *“Bivalirudin has a half-life of 25 min⁸ and must therefore be administered as a continuous infusion during PCI as the majority of these procedures last approximately 30 minutes to an hour. Variegin has a half-life of 52.3 ± 4.4 min³³ and may potentially be given as a single bolus for peri-PCI anticoagulation.”*

Methods:

- Which human study was approved (line 382)?

The IRB approval was for blood draw from healthy human volunteers for the global coagulation assays and thrombin generation tests.

- Provide synthesis and refolding protocols (if applicable) in greater detail, including the specific mass for each peptide compared to theoretical mass, after MS. What was the yield?

All peptides were synthesised using a standardized Fmoc-based solid phase synthesis method as implemented in CEM Liberty Blue automated peptide synthesizer, as described in the **“Peptide synthesis and purification”** section under **Materials and Methods** (Page 21 & 22). All peptides used are linear peptides without Cys, therefore no refolding was involved. As requested, we have now included the ESI mass spectra and deconvoluted mass spectra, observed and theoretical mass of each peptide in the Supplement (Supplementary fig. 1 to 14 and Supplementary Table 1). Regrettably, we do not routinely calculate chemical synthesis yield but all lyophilised peptides have been characterised with an analytical column for purity checks, consistent with synthesis reports typically provided by commercial vendors.

- Is the calculation of peptide concentration reliable, with no Tryptophan in the sequences (one Tyr is noted)? This is critical for the implications of the results, and need to be verified.

All but 2 of the peptides has one Tyr. We understand that the calculation based on absorbance and extinction coefficients at 280 nm based on Tyr may not be as reliable as Trp but the estimation is at least consistent when compared across this series of peptides with similar sequence and length and across different batches of synthesis. The concentrations of the 2 peptides without Tyr (Avathrin and UV004) were estimated using absorbance at 205 nm and extrapolated from standard curves constructed using peptides of identical length and similar sequences. We think all routinely accessible methods for protein/peptide concentration estimations have their own limitations. For example, in our experience, the dry weight of peptides typically underestimate peptide content due to the presence of acetate salt in lyophilised peptide. Fluorescent or calorimetric based methods require dyes, are indirect and are readily influenced by differences in peptide sequence. We have therefore selected the UV absorption method for its consistency. We also note that bivalirudin was synthesised, purified and has concentrations estimated using the same approach and was used as an internal quality and positive control in our experiments. For example, the K_i of bivalirudin was determined to be 1.8 ± 0.15 nM, and is reasonably consistent with the widely reported value of 2 nM, hence validating our approach.

We have therefore explained in the revised manuscript (Page 21-22, line 485-490):

“Concentrations of peptide solutions were estimated using UV absorbance at 280 nm and the extinction coefficient was calculated from peptide sequence. For the 2 peptides without Tyr (avathrin and UV004), we measured absorbance at 205 nm and estimated the concentrations based on standard curves constructed using peptides of identical length and similar sequences with known concentrations (eg. ultravariegin and UV003).”

- Rational for the CWA is required, not a universally used method.

We performed CWA as an extended interpretation of the data obtained from the aPTT (Fig. 3a). CWA has been increasingly adopted in research and clinical laboratories (*Wada et al. Clin Appl Thromb Hemost. 2020, 26:1076029620912027*). Of particular relevance to this project, the 1st derivative (min1) in CWA represents the effect of the “thrombin burst” and the absolute value of min1 is clinically associated with bleeding risk (*Suzuki et al. Thromb J. 2019. 17:12*). We have modified a sentence and added the relevant reference in Page 9, line 208-210, to add the rationale of performing CWA, as follows:

“The first derivative (min1) represents thrombin activity (ie. “thrombin burst”) and bleeding risk (low absolute value is associated with greater bleeding risk)³⁶”

Similarly, we have modified sentences in the **Discussion** to reflect the above (Page 18 & 19, line 418-422):

“...without impacting the rate of synthesis of various enzymes/complexes important to haemostasis, thus causing less bleeding (Fig. 3b). In contrast, 0.6 μM UFH and 2.7 μM bivalirudin resulted in severe impairment of coagulation (eg. min1 was less than half of control); a decrease in the absolute value for min1 is clinically associated with an increase bleeding risk (Fig. 3b)³⁶.”

- How many rats and pigs for each experiment/dose?

We have now included the number of animals used for each data point in the legends of Fig. 4, 5 and 6 (page 34-36). We have also replaced the bar charts used in these figures to scatter plots.

- Specify the source for pigs, i.e., were they in captivity, and if so where, no information is provided.

We have added the source for pigs in the “**Pigs**” section under **Materials and Methods** (Page 27, line 624-625):

“Both male and female SPF pigs (40-70 kg) were locally farmed for research purpose and obtained from the SEMC (SingHealth Experimental Medicine Centre, Singapore).”

- Explain why ACT was used, not PT or aPTT.

We used POC ACT monitoring to adjust the dose of UFH in pigs experiments as this is recommended by clinical practice guidelines for PCI (*Levine et al. J Am Coll Cardiol 2001, e44-e122*) (Page 29, line 659). In general, PT is insensitive to UFH and aPTT is not reliable for high or low doses of UFH (*Niccoli and Banning, Heart 2002, 331-334*).

- Bivalirudin recommended dose is 0.75 mg/Kg/hr in humans. How did the authors decide the dose of peptides to be used (0.025-0.25 mg/Kg)?

We did initial dosing trials in rats to determine the range of doses to be tested. The recommended dose of bivalirudin in human is an initial iv bolus of 0.75 mg/kg followed by continuous infusion of 1.75 mg/kg/h. The equivalent dose in rats for bivalirudin is calculated to be 4.65 mg/kg+10.85 mg/kg/h. From our experiments we realised that the clinically-approved doses of UFH and bivalirudin would result in maximum antithrombotic activity (ie. no occlusion within 60 min of observation). Considering the higher affinity of variegain and ultravariegain for thrombin (compared to bivalirudin), we initiated our dose-ranging trials with lower doses than for bivalirudin.

Results, Figures:

- Figure 2A – specify if the inhibitor (ultravariegain) behaves as a slow or fast tight type inhibitor.

We have modified line 152-154, Page 7, to indicate fast- and tight-binding behaviour of all peptides:

“...inhibitor and thrombin with linear inhibition progress curves (Fig. 2a). This is consistent with the previously reported fast- and tight-binding behavior of variegins-like peptides^{29,30}.”

- Figure 2C – the projection of the linear regression is negative when it intercepts the y axis (also indicate the Unit in the axis - pM? nM?), which can be used to calculate a K_i . How this reconcile with a K_i of 4 pM?

We thank the Review for astutely pointing out our error. Instead of fitting the data to the correct equation for competitive inhibition $K_i' = K_i (1 + S/K_m)$, we had used the default linear regression fitting function in Prism 6 to generate the straight line in Fig. 2c, resulting in this discrepancy. However, this did not affect the reported value of ultravariegin K_i (4.0 ± 0.5 pM) because we did not obtain the K_i graphically. Instead, the reported K_i was calculated as an average of 5 different K_i values obtained by solving the above equation at the 5 different substrate concentrations tested. In this revision, for the graphical representation in Fig. 2c, we have now used the line generated from fitting the data points to the equation $K_i' = K_i (1 + S/K_m)$ to rectify this problem. The y-intercept (K_i) of the new fitted line is 4.2 ± 0.3 pM, consistent with the reported K_i calculated as above. We have also added the unit for the y-axis (nM).

- Figure 2E – it is expected 100% inhibition of 0.83 nM thrombin with 1 nM of a bivalent tight inhibitor such as ultravariegin, and even more so with 10 nM. This is not the case. It might be that the concentration of the reactants is somewhat not accurate. Please verify.

As explained in response to the earlier comment on concentration estimation, we are confident that there should not be any outsized errors in the concentrations of the reactants. Even if there are inaccuracies in the concentration of reactants, the effect should be sufficiently overcome by the large excess in inhibitor concentrations (eg. 10 nM). Instead, we think one of the more likely reasons would be the presence of a small proportion of autolysed thrombin (eg. beta- and gamma-thrombin) in the preparation of thrombin used (purchased from Haemtech but shipped and stored in 50% glycerol solution at -20°C). Beta and gamma thrombins retains active site amidolytic functions (ie. hydrolysis of chromogenic substrate S2238) but are known to be less susceptible to inhibition by exosite-directing inhibitors (eg. *Ascenzi et al. J Mol Biol 1992, 177-184*). Again, since we used bivalirudin in our experiments as an internal positive control/validation, we do not believe that this issue affects our overall conclusions, given that the K_i value of bivalirudin in our hands is consistent with what is reported in the literature. We have added the source of thrombin in the revised manuscript (Page 22, line 495).

- Figure 3C – the actual plot for thrombin generation assay is needed, at least for ultravariegin.

We have added one set of thrombin generation plots for the four anticoagulants under all conditions as Supplemental figure 15.

- Figure 4D –Bivalirudin expressed in mg/Kg/h in the x axis because of continuous infusion. This makes any comparison more difficult as the peptides were tested in mg/Kg, as a bolus infusion. Did the author try bolus infusion of bivalirudin at similar concentration as other peptides so it can be compared? Please elaborate and discuss. Specifics regarding pharmacokinetics if available.

Since our aim for this project was to investigate variegins/ultravariegin for potential clinical use in peri-PCI anticoagulation, we have performed comparisons with bivalirudin the same way that it is being administered in the clinical setting, ie. through continuous infusion. As stated in the **Introduction**, we hypothesised that variegins with a half-life of 52.3 min may be ideal as a single bolus peri-PCI

anticoagulant as the majority of PCI procedures last approximately 30 – 60 min. In contrast, the short half-life of bivalirudin (25 min) would result in low antithrombotic efficacy if given as a bolus injection without a continuous infusion.

Earlier, we have tried a series of time-course experiments in rats to demonstrate this “high antithrombotic efficacy peri-PCI, low bleeding post-PCI” mode of action for variegien and compared it to bivalirudin/UFH. Clinically-approved dose of bivalirudin (10.8 mg/kg/h) were infused for 15 min and then turned off – *this would be the equivalent to the bolus infusion as suggested by the Reviewer*. Next we initiated carotid artery thrombosis and bleeding models at time = 5, 30, and 60 min after turning off the infusion. Bivalirudin was unable to effectively prolong the occlusion time after the infusion was turned off. This is consistent with human studies of bivalirudin in which the omission of a continuous infusion of bivalirudin after PCI led to an increased incidence of subacute stent thrombosis (Stone et al. N Engl J Med 2008, 2218-2230 and Valgimigli et al. N Engl J Med 2015, 997-1009). Eventually we decided that such experiments have little clinical relevance because this is not a valid/fair comparison with bivalirudin as it is approved for use as a continuous infusion clinically. Therefore, we did not continue with this series of experiments using ultravariegien and did not include the data in the submitted manuscript. However, we are including this information and the resulting plots here as a response to this comment.

- Figure 5 and 6 – it is difficult to really extrapolate the results with respect to bleeding vs antithrombotic activity with the concentrations tested, and imply safety of one molecule vs another. It could be that bivalirudin at 0.3 mg/Kg/hr shows the same antithrombotic effect at the tested concentration 0.75 mg/Kg, but would cause much less bleeding. Evidently, these experiments are exceptionally costly and time consuming for a dose-response curve, but few words in this respect are reassuring without diminishing the implications of authors` findings. How n in figure legends indicate number of animals per data point? Please clarify.

Since our aim was to investigate our novel DTIs as peri-PCI anticoagulants, we have chosen to use the dose of DAPT, UFH and bivalirudin to be consistent with their clinically-approved doses for PCI. As such we believe the data depicted in Fig. 5 & 6 need to be considered together. It showed that in the current setup without DAPT, UFH and bivalirudin at clinically-approved dose cannot fully prevent thrombus formation (ie. reduced by 38% and 61% compared to saline, respectively, Fig. 5b). In contrast, 0.25 mg/kg ultravariegien achieved better efficacy (95% thrombus reduction) than UFH and bivalirudin ($p \leq 0.0001$ & $p \leq 0.05$, respectively, Fig. 5b). Importantly, there is no significant difference between these groups for bleeding (Fig. 5c). Therefore, improvement of efficacy over clinically recommended doses of UFH and bivalirudin can be achieved with a high affinity inhibitor like ultravariegien, without increasing bleeding time compared with UFH and bivalirudin.

For UFH and bivalirudin, maximum efficacy can be achieved with clinically-approved doses of UFH and bivalirudin by combining them with DAPT (Fig. 6b) but this comes with a price of increased bleeding (Fig. 6c), and is consistent with results from clinical trials (Stone *et al. N Engl J Med* 2008, 2218-2230 and Valgimigli *et al. N Engl J Med* 2015, 997-1009). Since 0.25 mg/kg ultravariegin alone achieved high efficacy without DAPT (Fig. 5b), we reasoned when used in combination with DAPT, the dose of ultravariegin can be substantially lowered to reduce bleeding risk (Fig. 6c).

We agree with the Reviewer that dose optimisation for UFH or bivalirudin may result in better balance of efficacy:safety profiles for these drugs. However, we think there may not be enough room to robustly lower doses of UFH and bivalirudin without compromising their efficacy (eg. Fig5b). This hypothesis can also be supported by clinical evidence that bivalirudin is associated with increased stent thrombosis, and the effect can only be mitigated with a prolonged infusion of full, but not reduced, doses of bivalirudin post-PCI, as discussed in **Discussion** (Page 17-18, line 393-399). Extensive dose optimisation for all anticoagulants and antiplatelets in the porcine model at this point would be too resource-intensive and outside the scope of this project.

Therefore, we have stated in our discussion that *“Considering the important role of platelets in arterial thrombosis, this approach of combining a low-dose, high affinity anticoagulant with DAPT may achieve high antithrombotic efficacy while minimizing bleeding risk during PCI.”* (Page 18, line 409-411).

Further, in the discussion about limitations of our study, we have added this point:

“Third, we did not perform dose optimisation of UFH and bivalirudin when used together with DAPT, since we aimed to validate variegin/ultravariegin in comparisons with clinically-approved doses of UFH and bivalirudin. It remains possible that dose optimisation will result in a more balanced efficacy-to-bleeding profile of UFH or bivalirudin.” (Page 20, line 444-448)

Discussion

- Lines 355 – 359 – please elaborate why bivalirudin has a narrow therapeutic window compared to variegin?

Compared to ultravariegin, bivalirudin has a smaller window between occlusion and bleeding in rats (Fig. 4 and Supplementary Table 3), and more rapid increase in anticoagulant intensity over a narrower concentration range in thrombin generation experiments (Fig. 3, Supplementary Fig. 1 and Supplementary Table 1). These results showed that there would be a smaller dose range for optimum balance of efficacy and bleeding risks (smaller therapeutic window) for bivalirudin compared with ultravariegin.

We have added this sentence to the end of this paragraph to complete the discussion (Page 19, line 434-435):

“Furthermore, in our rodent models, variegin/ultravariegin also achieved a 3- to 7-fold wider therapeutic index compared with UFH/bivalirudin”

- Atherosclerotic plaque contain Tissue Factor, can the authors elaborate on strategies targeting TF in the therapy for PCI?

There has been limited success with TF-targeting drug in clinical development as an antithrombotic drug. We suspect TF may be a challenging target for indications such as PCI since it's role is confined within the initiation of coagulation and there are other signals that will lead to activation of platelets

and production of thrombin. Breakthrough production of thrombin can easily lead to the eventual thrombin burst. Hence, early and complete inhibition of TF may be required, which could be very challenging to achieve, especially in emergency PCI cases where patients frequently present to the cardiac catheterisation laboratory after rupture of an atherosclerotic plaque.

- Overall, this is an impressive work involving numerous areas of expertise that provides proof of concept to develop ultravariegin to the next step which is its potential clinical use. It also underlines the importance of the study of salivary components from blood-feeding arthropods as tools in biochemistry and in the development of novel therapeutics. Finally, please include your changes/replies to the reviewer, to the manuscript.

Once again, we thank the reviewer for the very helpful comments that we think have greatly improved our manuscript.

Reviewer #3 (Remarks to the Author):

As instructed, I have limited my peer-review to the ligands (peptides and antibodies) used.

We thank the reviewer for the comments. In our response, line numbers are based on the tracked change manuscript file.

During this revision, we realised that we had inadvertently used just one BIL dataset for K_d calculation in the initial submission. We have rectified that in the revision, plotting the steady state binding isotherms using the mean of both replicate datasets in Fig. 7b & c. The K_d values for ultravariegin binding to Ab1282 and Ab1283 are now corrected to 1.25 ± 0.07 nM (was 1.30 ± 0.23 nM) and 1.40 ± 0.42 nM (was 1.70 ± 0.93 nM) respectively (Page 14, Line 334-335).

I have two concerns:

1. There is no QA/QC for the synthetic peptides. As such it is challenging to determine purity of the preparation and rule out potential artifacts.

As instructed, we have now included the ESI mass spectra and deconvoluted mass spectra as well as the observed and theoretical mass of each peptides in Supplementary figures 1 to 14 for all peptides. A table of peptide name, sequence, observed and theoretical mass has also been included as Supplementary Table 1. The purity of all lyophilised peptides used for our assays were characterised with an analytical column for purity checks as indicated in the supplementary figures.

We have added these details in the manuscript (Page 21, line 479-485):

“Purification of peptides were performed on Jupiter® 4 µm Proteo 90Å (250 x 21.2 mm) reversed-phase column (Phenomenex, CA, USA). The purity and masses of peptides were determined on Aeris™ 3.6 µm Widespore XB-C18 100 Å (150 x 4.6 mm) reversed-phase column (Phenomenex, CA, USA) and by electrospray ionization mass spectrometry (ESI-MS) using an LCQ Fleet Ion Trap MS (Thermo Fisher Scientific, MA, USA) (Supplementary Fig. 1 to 14 and Supplementary Table 1).”

2. While the authors appropriately "pre-empt" the criticism that the seven monoclonal antibodies selected from a naive phage library were only tested in vitro in the end of the manuscript Discussion, instead they should really test at least one of their leading candidates in vivo and demonstrate reversal of their DTIs.

For the revision, we have expressed and purified more Ab1283 for animal experiments, although producing even larger quantities remained challenging. We have added the following in the **“Rat tail incision bleeding model”** under **Materials and Methods** for *in vivo* validation of Ab1283 (Page 27, line 607-610):

“This model was also used for testing of in vivo reversal activity of Ab1283. Four minutes after rats were injected with an i.v. bolus of 0.75 mg/kg ultravariegin, an i.v. bolus of 10 mg of Ab1283 or saline was injected. One minute later, an incision was made on the tail and bleeding time was measured as described above.”

In the **“Reversal agents for ultravariegin were identified from a naïve human antibody library”** section of **Results**, the description of the results is now added (Page 15, line 338-341):

“Next, reversal activity of Ab1283 was tested in vivo. Rats that were injected with 0.75 mg/kg ultravariegin and followed by 10 mg of Ab1283 showed a statistically significant reduction in tail

bleeding time (12.7 min vs 16.8 min, $P \leq 0.01$) compared to those receiving saline in place of Ab1283 (Fig. 7d)."

The reversal of ultravariegin activity *in vivo* is statistically significant but incomplete. However, there are a few considerations here. Ab1283 was expressed as an IgG antibody with molecular weight of 150 kDa, around 44X higher than ultravariegin. Therefore, the dose administered (0.75 mg/kg ultravariegin vs around 33 mg/kg Ab1283) assumes 1:1 binding. However, the affinity of ultravariegin for thrombin is unusually high (4 pM). Although affinity between ultravariegin and Ab1283 is in the low nanomolar range (1.4 nM), it would be challenging for Ab1283 to remove thrombin-bound ultravariegin. Further, given the smaller size of ultravariegin, distribution into the extravascular space is likely while Ab1283 is confined in blood. Indeed, pharmacokinetic studies of variegin have shown that the volume of distribution is around 750 ml/kg in pig (an adult pig's blood volume is around 65 ml/kg, Shih et al, Bioanalysis 2017), indicating significant extravascular distribution. All these factors potentially contributed to incomplete reversal of ultravariegin's anticoagulant activity in our experiments. Notably, dabigatran's clinically-approved antidote (idarucizumab, Fab, MW = 47.8kDa) similarly requires very large doses for effective reversal (5 g) (Pollack et al. *N Engl J Med* 2017, 431-441). Therefore, we think that Ab1283 is a promising starting point for antidote development.

With this set of experiments, we have also added the following to the **Discussions**:

Page 16, line 365-368: *"Finally, two specific reversal agents showing nanomolar level binding affinity to ultravariegin were identified. In vivo reversal experiments of rats anticoagulated with ultravariegin demonstrated that Ab1283 reduced bleeding time compared with control, representing a potential lead for antidote development."*

Page 19 & 20, line 448-451: *"Fourth, although the two antibodies were able to fully reverse the thrombin inhibition effect of ultravariegin in vitro, the in vivo studies showed that large doses of Ab1283 may be required for complete reversal, hence the molecule may require further optimisation as an antidote."*

Reviewer #4 (Remarks to the Author):

Koh and co-workers have characterized potential thrombin-inhibiting peptides present in tick saliva and then optimized amino acids from one of these peptides, variegins, to produce a final peptide called ultravariagin. Ultravariagin inhibits thrombin with a K_i of 4 pM but has little affinity for other coagulation proteases. The anticoagulant activity of variegins and ultravariagin were compared to UFH and bivalirudin in a series of coagulation assays performed in human plasma or human platelet rich plasma. Their ability to inhibit clotting and produce bleeding were examined using *in vivo/ex vivo* assays in rats and pigs. Ultravariagin reduced thrombin formation more gradually than UFH or bivalirudin in the plasma clotting assays. In animal experiments, ultravariagin prevented thrombosis while producing much less bleeding than UFH or bivalirudin (Figures 6B & C). The findings are of interest because these pre-clinical data suggest that these tick-derived peptides may have much better efficacy to safety profiles than previously developed direct thrombin inhibitors. Thus, these studies advance the field for further development of this class of anticoagulants for use in percutaneous coronary interventions. I have the following comments for the authors.

We thank the reviewer for these helpful comments. In our response, line numbers are based on the tracked change manuscript file.

1. The thrombin generation data indicate that ultravariagin inhibits thrombin formation more gradually as its concentration is increased when compared to bivalirudin with the largest differences in the change in lag time and change in time to peak thrombin. It is concluded that this gradual inhibition preserves a capacity to regenerate thrombin for haemostasis when needed, and potentially explains why ultravariagin has better antithrombotic efficacy with less bleeding than UFH or bivalirudin. This conclusion deserves further explanation from the authors.
a. How does the mechanism inhibition of thrombin by ultravariagin differ from that of bivalirudin that accounts for the unique effects of ultravariagin?

There are a number of major differences between ultravariagin and bivalirudin. These include the higher affinity of ultravariagin for thrombin (445-fold better than bivalirudin) and the important structural differences in the amino acid sequence, especially the C-terminal aspect of the thrombin cleavage scissile bond, as discussed in Page 17, line 385-390: *“Several structural-functional properties of variegins-like peptides may account for these differences from bivalirudin in pharmacokinetic-pharmacodynamic relationships. For example, variegins-like peptides contain specific amino acid residues within the active site and extended amino- and carboxyl-termini leading to stronger binding of thrombin. In contrast, bivalirudin has a non-specific, flexible tetra-glycyl linker between the active site and exosite-I binding sequence, resulting in loss of activity upon cleavage by thrombin^{29,30,31,32}.”*

Thrombin cleaves bivalirudin between Arg3-Pro4 upon binding. We have previously demonstrated that thrombin also cleaves variegins between the Lys10-His11 scissile bond; the cleavage product C-terminal to the scissile bond (sequence: MHKTAPPDFEAIPEEYLDDDES) was named MH22 and was found to be a non-competitive inhibitor of the thrombin active site with K_i around 14 nM (Koh *et al*, *Chembiochem* 2009, 2155-2158). In this revision, we synthesized two new peptides: UV011 and BV001. We have further demonstrated that the equivalent cleavage product of ultravariagin (sequence: MYSTAPPDGFEEIPDDAIEE), named UV011, can also inhibit thrombin with a K_i of 1.66 ± 0.76 nM (Fig. 2d and Table 1), assuming the same non-competitive mode of inhibition as MH22. We have also demonstrated that the equivalent cleavage product of bivalirudin (sequence PGGGGNGDFEEIPEEYL), named BV001, is unable to inhibit thrombin but instead leads to 20% activation of the thrombin active site when tested at concentrations higher than 1 μ M. The difference between continual, albeit weaker, inhibition of thrombin by ultravariagin despite cleavage and the total loss of inhibition by bivalirudin

upon cleavage could account for the unique mechanism of inhibition by ultravariegin. We have added these results as Fig 2d, added the sequence of UV011 and BV001 to Table 1 and have added the following to the manuscript:

Page 8 & 9, line 182-191: *“We have previously reported that thrombin cleaves variegin between the Lys10-His11 scissile bond but the cleavage product C-terminal to the scissile bond (sequence: MHKTAPPDFEAIPEEYLDDDES) is a non-competitive inhibitor of thrombin’s active site with a K_i of ~ 14.1 nM³¹. We synthesised the equivalent cleavage product of ultravariegin UV011 and showed that it inhibits thrombin (Table 1 and Fig. 2d). Assuming the same non-competitive mode of inhibition, the K_i of UV011 is 1.66 ± 0.76 nM. Bivalirudin is also cleaved by thrombin upon binding^{31,35} and we synthesized the equivalent cleavage product of bivalirudin (BV001) (Table 1). Surprisingly, BV001 was not only unable to inhibit thrombin but instead activate thrombin’s active site by up to 20% when tested at concentrations higher than 1 μ M (Fig. 2d).”*

Page 17, line 391-393: *“Using a pair of peptides, UV011 and BV001, we have demonstrated continual, albeit weaker inhibition of thrombin after cleavage of ultravariegin (Fig. 2d), in contrast to complete loss of inhibition by bivalirudin after cleavage (Fig. 2d).”*

We have several theories regarding why variegin/ultravariegin showed more gradual inhibition of thrombin with increasing concentrations compared with bivalirudin. In *in vitro* assays (eg. thrombin generation assays), cleavage of ultravariegin results in two species of thrombin inhibitors: ultravariegin with higher affinity ($K_i = 4.0$ pM) and UV011 with 415-fold weaker affinity ($K_i = 1660$ pM). UV011 may ‘compete’ with ultravariegin for thrombin binding and hence attenuate the overall observed inhibition in this reaction. Accordingly, increasing ultravariegin concentration does not result in as strong an increase in anticoagulation intensity as otherwise expected. In contrast, bivalirudin’s cleavage product does not bind to thrombin, and any increase or decrease in the concentration of bivalirudin is fully reflected in its anticoagulation intensity. However, we understand much of the above remains hypothetical and we have adopted a cautious approach to explaining this hypothesis in the **Discussion** (Page 19, line 427-435).

b. Why does inhibition of thrombin generation by ultravariegin allow for regeneration of thrombin in a tail clip or ear puncture wound but not on a stent or de-endothelialized vasculature.

We postulate that because ultravariegin demonstrates a more gradual increase in anticoagulation response over a wider concentration range compared with bivalirudin, it enables a wide therapeutic window that produces an optimum balance between antithrombotic efficacy and bleeding. The window can be further widened using our strategy of a low dose, high affinity DTI (ie ultravariegin) in combination with antiplatelet therapy (DAPT). For arterial thrombosis (eg. PCI) in which platelet activity plays a dominant role, the use of DAPT allows micro-dosing of ultravariegin. In the event of bleeding (eg. tail or ear vein bleeding), it would be much easier for haemostatic processes to overcome the tiny dose of ultravariegin. In contrast, bivalirudin’s low affinity for thrombin does not allow for a similar low-dose strategy when used in combination with DAPT.

We have modified part of our discussion to bring attention to this point above (Page 18, line 409-411):

“Considering the important role of platelets in arterial thrombosis, this approach of combining a low-dose, high affinity anticoagulant with DAPT may achieve high antithrombotic efficacy while minimizing bleeding risk during PCI.”

2. A concern with all bleeding models in animals is that they poorly predict actual bleeding in humans.

The data are compelling that in the models studied ultravariegen produces less bleeding than UFH or bivalirudin. Nevertheless, additional discussion of the weaknesses of the two bleeding models used is warranted in the limitations paragraph in the discussion.

We agree with the Reviewer that there is a lack of good bleeding models in animals that accurately predict clinical bleeding risk. In collaboration with Maureane Hoffman and Dougald Monroe, we have now performed additional experiments using a more technically-challenging reference standard bleeding model – the saphenous vein bleeding model (*Monroe and Hoffman Thromb Res 2014, S6-S8*) - and showed that the results obtained correlate well with those from the tail bleeding model (Fig. 4e & f). Methods for the rat saphenous vein bleeding model have been added to Page 27, line 611-622. Results for the model have been added to Page 12, line 274-284. We have also added the following sentences to the discussion in the limitations of our study (Page 19 & 20, line 441-444):

“Pre-clinical animal bleeding models may not always predict clinical bleeding risks in human trials because of a variety of reasons^{53,54}. Here, we used three different bleeding models (tail, saphenous vein and superficial ear vein) to increase confidence in the results.”

REVIEWERS' COMMENTS

Reviewer #1 (Remarks to the Author):

Thanks to the authors for their responses to my comments and suggestions. I am satisfied with their responses.

Reviewer #2 (Remarks to the Author):

Authors have answered all questions in great detail, and changed the manuscript accordingly. Nice study.

Reviewer #3 (Remarks to the Author):

The authors have made a bona fide effort to address my two criticisms.

Reviewer #4 (Remarks to the Author):

The authors have addressed my concerns in the revised version.

REVIEWERS' COMMENTS

Reviewer #1 (Remarks to the Author):

Thanks to the authors for their responses to my comments and suggestions. I am satisfied with their responses.

Reviewer #2 (Remarks to the Author):

Authors have answered all questions in great detail, and changed the manuscript accordingly. Nice study.

Reviewer #3 (Remarks to the Author):

The authors have made a bona fide effort to address my two criticisms.

Reviewer #4 (Remarks to the Author):

The authors have addressed my concerns in the revised version.

RESPONSE

We would like to thank all reviewers for their helpful comments. We note that all comments are positive and have no further request for changes.